# Lavida-O: Elastic Large Masked Diffusion Models for Unified Multimodal Understanding and Generation

**Shufan Li**[1,2,*], **Jiuxiang Gu**[1], **Kangning Liu**[1], **Zhe Lin**[1], **Zijun Wei**[1]
**Aditya Grover**[2], **Jason Kuen**[1]
[1]Adobe  [2]UCLA
* Work done primarily during internship at Adobe Research

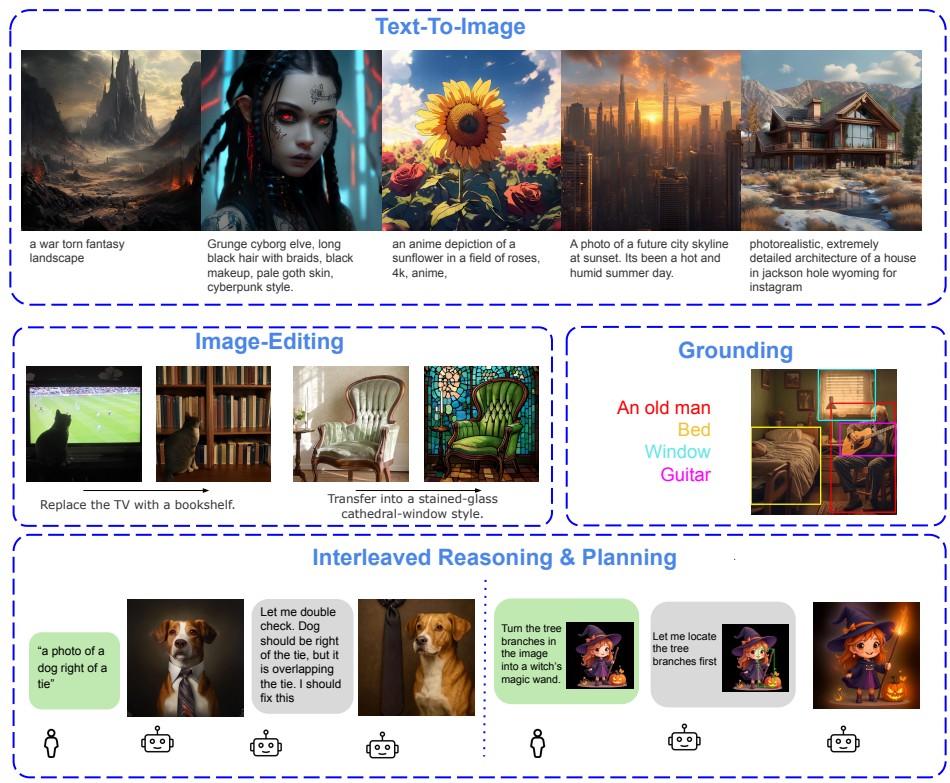

Figure 1: **We propose Lavida-O,** a unified large masked diffusion model capable of multi-modal understanding and generation.

## Abstract

We propose Lavida-O, a unified Masked Diffusion Model (MDM) for multi-modal understanding and generation. Unlike existing multimodal MDMs such as MMaDa and Muddit which only support simple image-level understanding tasks and low-resolution image generation, Lavida-O presents a single framework that enables image-level understanding, object grounding, image editing, and high-resolution (1024px) text-to-image synthesis. Lavida-O incorporates a novel Elastic Mixture-of-Transformers (Elastic-MoT) architecture that couples a lightweight generation branch with a larger understanding branch, supported by token compression, universal text conditioning and stratified sampling for efficient and high-quality generation. Lavida-O further incorporates planning and iterative self-reflection in image generation and editing tasks, seamlessly boosting generation quality with its understanding capabilities. Lavida-O achieves state-of-the-art

performance on a wide range of benchmarks including RefCOCO object grounding, GenEval text-to-image generation, and ImgEdit image editing, outperforming existing autoregressive models and continuous diffusion models such as Qwen2.5-VL and FluxKontext-dev, while offering substantial speedup at inference. These advances establish Lavida-O as a new paradigm for scalable multimodal reasoning and generation. Code and checkpoints are available at https://github.com/adobe-research/LaVida-O.

# 1 INTRODUCTION

The abilities to understand and generate images have been two essential objectives of image modeling research. Traditionally, these tasks are handled by a diverse set of specialist models, such as detection models for object localization (Liu et al., 2024b; Li et al., 2023a), Visual Question Answering (VQA) models for question-answering (Li et al., 2022), and diffusion models for text-to-image generation (Esser et al., 2024; Podell et al., 2023; Rombach et al., 2022). Recently, the rise of unified multi-modal models such as GPT-4o (OpenAI, 2024) has introduced a new paradigm: using a single generalist model to perform a wide range of image understanding and generation tasks. Not only is this unified approach more aligned with the goal of developing versatile multi-task Artificial General Intelligence (AGI), but it also demonstrates strong empirical performance by allowing understanding and generation capabilities to mutually benefit each other (Deng et al., 2025). This is especially notable in tasks requiring both understanding and generation capabilities, such as image editing, where unified models show unparalleled advantages over generation specialists.

Most current unified models are built on Autoregressive (AR) large language models. Some works, such as BLIP3o (Chen et al., 2025a) and BAGEL (Deng et al., 2025), employ AR modeling for text generation and continuous diffusion modeling for image generation (AR+diff), while others, such as Janus (Chen et al., 2025c), first tokenize images into sequences of discrete tokens and then employ a unified AR next-token prediction objective for both image and text modalities.

Recently, Masked Diffusion Models (MDMs) (Lou et al., 2023; Sahoo et al., 2024) have emerged as a competitive alternative to AR models. Unlike AR models, MDMs treat token generation as a diffusion process over discrete tokens. In the forward process, the tokens of a sequence are gradually masked. At inference, we start with a sequence of mask tokens and gradually unmask them to obtain a sequence of meaningful tokens. Large-scale experiments in language modeling (Nie et al., 2025; Ye et al., 2025a) show that MDMs can achieve comparable performance to AR language models while offering many advantages, such as better speed-quality tradeoffs, controllability, and bidirectional context. Several recent works extend MDMs to multi-modal understanding and generation tasks (Li et al., 2025a; Yu et al., 2025b; Yang et al., 2025; Shi et al., 2025). Compared with the AR+diff setup, unified MDMs avoid the need to carefully tune the balance between AR and diffusion losses by offering a unified objective, resulting in greater simplicity and scalability. Compared with unified AR modeling, unified MDMs offer significantly faster sampling speeds by allowing parallel decoding of multiple tokens.

Despite these advantages, the latest unified MDMs—such as MMaDa (Yang et al., 2025) and Muddit (Shi et al., 2025)—still lag behind state-of-the-art unified AR and AR+diffusion models, both in the breadth of tasks they support and in benchmark performance. There are three main challenges in developing high-performing unified MDMs. First, unified models are expensive to train due to the large size of their language backbones. For example, to build a unified MDM with image generation capability, MMaDa pretrains an 8B model jointly on text and image generation, which is costly. This challenge is further exacerbated by the limited literature on training large-scale masked image generative models. In contrast, many open-source large-scale continuous diffusion models such as Flux (Labs, 2024) are readily available. Second, open-source resources for masked image generative models (MIGMs) are scarce, and the literature on their training techniques and sampling processes is less developed than that for continuous diffusion models. Even the best open-source MIGM, Meissonic-1B (Bai et al., 2024), significantly underperforms continuous diffusion models of comparable size (Xie et al., 2025a). Lastly, while these models can perform both understanding and generation tasks, they lack explicit mechanisms to leverage image understanding capabilities to improve generation quality. In fact, MMaDa and Muddit cannot even perform image editing tasks,

which require both understanding and generation capabilities. These models simply concatenate text-to-image data and image understanding data during training.

To bridge this gap, we propose Lavida-O, a unified multi-modal Masked Diffusion Model (MDM) capable of both image understanding and generation tasks. To mitigate the cost of training large diffusion models, Lavida-O introduces several techniques such as Elastic Mixture-of-Transformers (Elastic-MoT), progressive upscaling (gradually increasing the image resolution during training), and token compression that enable efficient scaling. To improve generation quality, Lavida-O employs stratified sampling and universal text conditioning. To fully leverage the potential of a unified multi-modal model, Lavida-O incorporates planning and self-reflection mechanisms that explicitly utilize its understanding capabilities to enhance generation outputs. We highlight Lavida-O's capabilities compared with previous multi-modal MDMs in Table 1.

Through extensive experiments, we show that Lavida-O achieves state-of-the-art performance on a wide range of benchmarks such as RefCOCO object grounding (Kazemzadeh et al., 2014), GenEval text-to-image generation (Ghosh et al., 2023), and ImgEdit (Ye et al., 2025b) image editing, outperforming existing autoregressive and continuous diffusion models such as Qwen2.5-VL (Bai et al., 2025) and Flux .1 Kontext dev (Labs et al., 2025), while offering up to a $6.8\times$ speedup. Overall, our contributions can be summarized as follows:

- We propose the first multi-modal MDM that achieves state-of-the-art performance on text-to-image generation, image editing, and grounding tasks, outperforming existing MDMs, AR models, and continuous diffusion models.

- We propose several efficient and effective training and inference techniques for large-scale masked image generative models and unified multi-modal models, such as the Elastic-MoT architecture, universal text conditioning, and stratified sampling, significantly advancing the literature.

- We introduce a novel paradigm that explicitly leverages the understanding capabilities of a unified model to improve its generation through planning and self-reflection.

Table 1: **Capabilities of different multimodal MDMs.** Lavida-O uniquely supports localized understanding, high-resolution image synthesis, image editing and interleaved generation.

| Model | Understanding | | Generation | | |
| --- | --- | --- | --- | --- | --- |
| | Image-level | Object-level | Text-to-image | Image-editing | Interleaved |
| LaViDa, Dimple, LLaDa-V | ✓ | × | × | × | × |
| Muddit | ✓ | × | $512^2$ | $*^1$ | × |
| MMaDa | ✓ | × | $512^2$ | × | × |
| LaViDa-O | ✓ | ✓ | $1024^2$ | ✓ | ✓ |

## 2 BACKGROUND AND RELATED WORKS

### 2.1 MASKED DIFFUSION MODELS

Masked Generative Modeling (MGM) has emerged as an alternative to AR models for modeling sequences of discrete tokens. Early works such as BERT (Devlin et al., 2019) used MGM as a representation learning objective. Later works (Chang et al., 2022; 2023) such as MaskGIT explored using MGM for generative modeling. In this setup, a sequence is initialized with only mask tokens, which are then gradually unmasked to generate the desired output. In these works, discrete tokenizers like VQGAN (Esser et al., 2021) are used to convert images into discrete tokens.

More recently, MDMs (Austin et al., 2021; Sahoo et al., 2024; Lou et al., 2023) have further developed the theory of MGM by formalizing the masking and unmasking process as the forward and reverse diffusion processes in discrete space. This provides a principled framework for training and

---

[1]Muddit showed examples of simple editing through inpainting. It does not have instruction-based editing capabilities.

sampling from these models. MDMs have renewed interest in masked modeling for language generation, offering theoretical advantages over AR models, such as better speed-quality tradeoffs and improved controllability. Notably, LLaDa-8B and Dream-8B (Nie et al., 2025; Ye et al., 2025a) demonstrated that MDMs can achieve competitive performance compared to AR models at scale. Several follow-up works (Li et al., 2025a; Yu et al., 2025b; You et al., 2025; Yang et al., 2025) such as LaViDa extend MDMs to multi-modal tasks such as image understanding and text-to-image generation. Their capabilities are summarized in Table 1.

Formally, given a sequence of $L$ discrete tokens $X_0 = [X_0^1, X_0^2, \ldots, X_0^L]$, the forward process $q(X_t|X_s)$ gradually masks the tokens over the time interval [0,1], with $1 \geq t \geq s \geq 0$. At $t = 1$, the sequence $X_1$ consists entirely of masked tokens, denoted by $[M]$. A neural network $p_\theta$ is used to model the reverse process $p(X_s|X_t)$. The masked diffusion objective is defined as:

$$\mathcal{L}_{\text{MDM}} = -\mathbb{E}_{t,X_0,X_t} \left[ \frac{1}{t} \log p_\theta(X_0|X_t) \right] \tag{1}$$

where $p_\theta(X_0|X_t)$ is factorized into $\prod_{i=1}^{L} p_\theta(X_0^i|X_t)$ based on independence assumptions (Sahoo et al., 2024). At inference time, the model starts from a fully masked sequence $X_1 = [M, M, \ldots, M]$ and progressively applies the learned reverse process $\log p_\theta(X_0|X_t)$ to recover the original tokens. We provide a more detailed formulation of MDMs in Appendix A.1.

## 2.2 UNIFIED MULTI-MODAL MODELS

Unified multi-modal models such as GPT-4o (OpenAI, 2024) are capable of both image understanding and generation tasks, leading to strong performance on tasks requiring both capabilities, such as image editing. Generally, there are two dominant types of unified models based on their modeling objectives. The first type, such as BAGEL (Deng et al., 2025), employs an AR objective for text generation and a diffusion objective for image generation (AR+diff). However, this design involves two different training objectives with distinct numerical scales and training dynamics, often requiring careful tuning of loss weighting and data mixtures. In contrast, the second type of models employ a unified objectives for both image and texts. Early works like Janus-Pro (Chen et al., 2025c) employ a unified AR modeling objective. Recent works like UMDD and MMaDa (Yang et al., 2025; Wang & Shi, 2008) explore a unified MDM objective. Despite some success, a significant performance gap remains between these unified MDMs and state-of-the-art unified models in the AR and AR+diff categories.

Architecturally, unified models also fall into two main categories. The first type, such as Janus and MMaDa, uses a single dense transformer to output both image and text tokens. The second type, such as BAGEL and MetaQueries (Pan et al., 2025), employs separate parameter sets for handling image and text modalities. A common design in this category is the mixture-of-transformers (MoT) architecture (Liang et al., 2024), where image and text inputs are processed by different parameter sets but can interact through joint attention mechanisms. Under this paradigm, several works such as X-Fusion and LM-Fusion (Mo et al., 2025; Shi et al., 2024) further investigated architecture designs and training recipes of MoT. These designs are illustrated in Figure 3. While being more flexible, training MoT experts can be expensive due to their large parameter counts.

## 3 METHOD

### 3.1 MODEL ARCHITECTURE

Lavida-O's model architecture is built on LaViDa (Li et al., 2025a), a diffusion model capable of only image understanding tasks. LaViDa uses a SigLIP (Zhai et al., 2023) vision encoder to convert input images into continuous semantic embeddings $C_i$, which are concatenated with token embeddings of text prompts $C_t$ to form the final conditional embeddings $C = \text{Concat}(C_i, C_p)$ for visual understanding tasks. At each inference step, the diffusion model uses the partially unmasked answer $X_t$ and the conditional embedding $C$ to predict the clean text answer $X_0$.

For image understanding tasks, Lavida-O maintains this exact setup of LaViDa. To incorporate visual generation tasks, we extend LaViDa's design by representing target images as sequences of

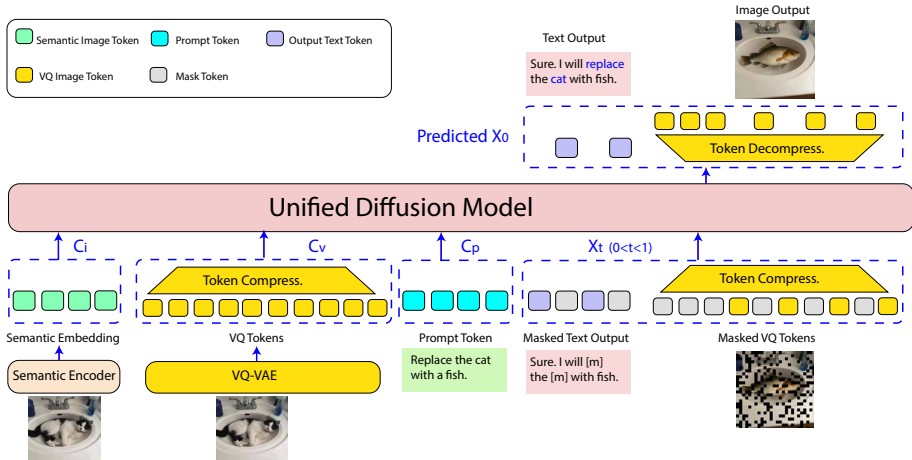

Figure 2: **Overall Pipeline of Lavida-O.** Given an input image and text prompt, we first concatenate the image semantic embedding $C_i$, image VQ embedding $C_v$, and text prompt embedding $C_p$ to form the conditioning embedding $C$. The combined embedding is then passed to the model alongside the partially masked sequence $X_t$. The model then predicts the fully-unmasked sequence $X_0$.

discrete tokens using a VQ-Encoder (Esser et al., 2021). When performing these tasks, $X_0$ and $X_t$ contain not only text tokens, but also VQ tokens that represent images. For image editing and interleaved generation tasks, we additionally incorporate VQ tokens of input images $C_v$ as part of the conditional embedding $C = \text{Concat}(C_i, C_v, C_p)$, since using semantic embeddings $C_i$ alone can degrade the low-level details needed for editing. To reduce the number of tokens and improve computational efficiency, we introduce a token compression module that reduces the number of VQ tokens by a factor of 4. The overall pipeline is illustrated in Figure 2.

### 3.1.1 ELASTIC MIXTURE-OF-TRANSFORMERS (ELASTICMOT)

Our goal is to find an efficient method that can equip an understanding-only diffusion model with visual generation capabilities. However, both of the existing common choices described in Section 2.2—dense models and MoT—are very expensive. Dense models use the same set of parameters for all tasks, requiring a mix of understanding and generation data during training to prevent catastrophic forgetting, which is not data-efficient. While the MoT setup allows freezing the understanding branch and training only the generation branch for image generation, its architecture doubles the total parameter count, leading to considerable computational overhead. Moreover, given an 8B base understanding model, both setups require training at least 8B parameters for generation tasks from scratch, which is prohibitively expensive.

To address these limitations, we propose Elastic-MoT, a novel architecture design that efficiently adapts an understanding-only model for image generation tasks. Compared with the vanilla MoT architecture, Elastic-MoT introduces two major modifications. First, instead of using equally sized branches, the generation branch has a smaller hidden size. This reduces the parameter count and enables efficient training. We make this design choice based on the observation that many text-to-image models can generate high-quality images with only 2–4B parameters, suggesting that generation tasks may not require as much capacity as understanding tasks (Xie et al., 2025a;b).

Second, given an $N$-layer model, instead of having joint attention at all layers, we only allow text and image modalities to interact in the first $M$ layers. In the remaining $K = N - M$ layers, text and image tokens interact only within their modality through self-attention. This design activates only partial parameters for different tasks. For example, in Lavida-O's final design, the generation branch has 2.4B new parameters and the understanding branch 8B parameters from LaViDa. With $N = 32$ layers and $M = K = 16$, image generation activates only 6.4B parameters (2.4B from generation + 4B from the first 16 understanding layers). During text-to-image pretraining, only the 2.4B generation branch is trainable, further improving the efficiency. Similarly, understanding

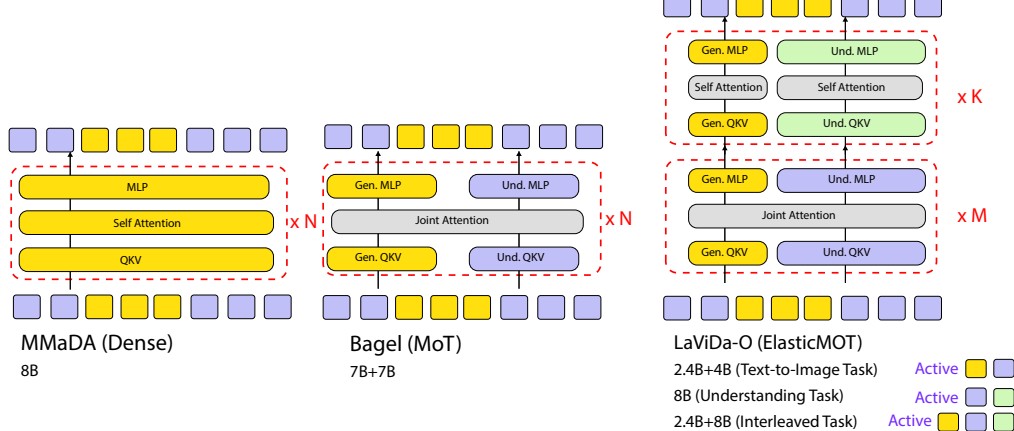

Figure 3: **Design of Elastic MoT.** Elastic-MoT introduces two major modifications to standard MoT. First, the generation branch has a smaller hidden size. Second, given an $N$-layer model, we only allow text and image modalities to interact in the first $M$ layers. These two designs allow us to flexibly load only a portion of parameters depending on tasks, improving the efficiency.

tasks use 8B active parameters, while interleaved tasks requiring both branches use 10.4B. The full Elastic-MoT design is shown in Figure 3, with further details in Appendix A.2 and B.2.

### 3.1.2 MODALITY-AWARE MASKING

One of the challenges in adapting MoT architecture for MDMs is routing—the mechanism to determine which branch should be activated for each token. This is trivial for unified AR MoT models, where the model can simply learn to generate a special token (e.g., [img_start]) to indicate that the next token should use the generation branch. However, MDMs decode tokens in parallel and must decide in advance which mask tokens should be routed to the understanding branch and which to the generation branch. A naive solution is to let the user specify the number and location of text and image tokens, but this is difficult for interleaved generation, such as image generation with self-reflection. To address this issue, we design a modality-aware masking process.

Given a sequence of $M$ text tokens and $N$ image VQ tokens, the vanilla forward diffusion process gradually converts it into $M + N$ mask tokens during the time interval $[0, 1]$. By contrast, our modality-aware forward process introduces a special timestamp $t_{exp} \in [0, 1]$, at which fully masked image VQ tokens are collapsed into a special [exp] text token. This process is illustrated in Figure 4a (Bottom-up). At inference, we assume all mask tokens are text tokens at the beginning. When a [exp] token is generated, we replace it with a sequence of $L_{img}$ mask tokens, and specify that these tokens will be processed by the generation branch for image synthesis in subsequent forward calls. This process is also illustrated in Figure 4a (Top-down). We provide additional details in Appendix A.3.

### 3.2 TASK-SPECIFIC DESIGNS

In this section, we describe several additional technical innovations that improve the effectiveness and efficiency on newly incorporated tasks such as image generation, image editing and grounding.

**Universal Text Conditioning**. A common approach to improving the quality of text-to-image models is micro-conditioning (Podell et al., 2023), which conditions the image generation process on extra parameters such as original image resolution, crop coordinates, and image quality scores. This is typically achieved via specialized embeddings. However, since a unified model has strong language understanding and reasoning capabilities, we can simply append these conditions as plain text (e.g., "SCORE: 5.40") to the end of user prompts. In addition to common conditions, we also incorporate image luminance and contrast as micro-conditions. This simple and effective design not

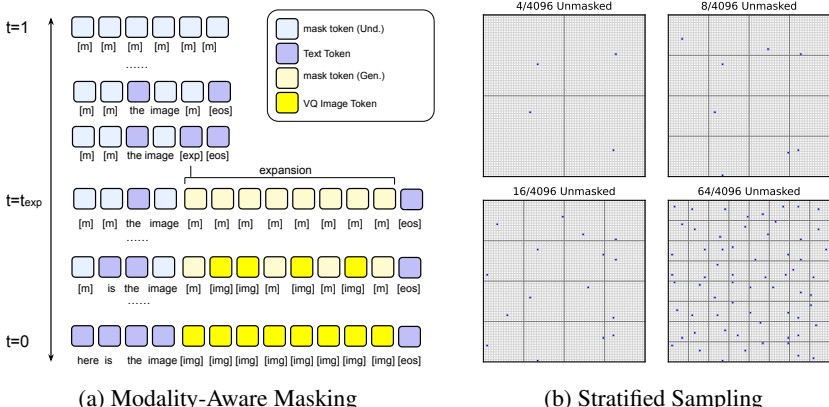

(a) Modality-Aware Masking        (b) Stratified Sampling

Figure 4: **Design choices of Lavida-O**. (a) Forward diffusion process with modality-aware masking. (b) Visualization of the unmasking order in the proposed stratified random sampling process.

only improves image quality by biasing generation toward high-scoring distributions, but also gives users more refined control over outputs. We provide additional details in Appendix A.4.

**Stratified Random Sampling.** Most MDMs use confidence-based sampling, unmasking high-confidence tokens first. In image generation, high-confidence tokens tend to cluster around already unmasked tokens. This negatively affecting image quality because adjacent tokens are highly correlated, which contradicts the independence assumption of MDMs. To mitigate this, we introduced a stratified sampling process. Starting with a $2 \times 2$ grid, we unmask one token per region to ensure broad spatial coverage. Each region is then recursively subdivided into four smaller subregions, and we continue unmasking one token from each new region. This process repeats until all tokens are revealed, producing a balanced, evenly distributed unmasking pattern across the entire image. This is illustrated in Figure 4b. More details and analysis are provided in Appendix A.5 and B.3.

**Planning and Reflection.** While existing unified MDMs integrate image understanding and generation tasks with a single objective, they do not incorporate mechanisms that use understanding to improve generation, except for the assumption that joint training benefits both tasks. To address this, we introduce two explicit mechanisms that leverage understanding to improve generation: *planning* and *reflection*. With planning, the model first generates a layout of the image represented by bounding boxes, then creates the actual image accordingly. For image editing tasks, it first identifies the desired edit region before generating the edited image. With reflection, the model evaluates its own generation using its understanding capability and determines whether it satisfies the user's request. If misalignment is detected, the model generates a new image correcting the error. Examples are shown in Figure 1, with additional technical results and analysis in Appendix A.7 and B.5.

**Object Grounding with Coordinate Quantization**. The bi-directional context of MDMs naturally allows parallel decoding of bounding box coordinates. While Lavida-O can represent numbers as plain text, we adopt a specialized scheme that normalizes all bounding box coordinates to $[0, 1]$ and quantizes them into 1025 discrete tokens representing $\frac{0}{1024}, \frac{1}{1024}, ..., \frac{1024}{1024}$. This ensures each bounding box is represented by exactly four tokens. At inference, we construct a multiple query input with masked tokens such as "A dog [m][m][m][m]; A cat [m][m][m][m]", and unmask all coordinates in parallel. This design allow us to decode multiple bounding boxes in as low as a single diffusion step, greatly boosting the efficiency. We provide further details in Appendix A.6

## 4 EXPERIMENT

### 4.1 SETUP

We start with LaViDa (Li et al., 2025a) and extend it with a 2.4B image generation branch using the ElasticMoT architecture described in Section 3.1.1. The training consists of three stages: **Stage 1:** We continue training the base model on object grounding and image-level understanding tasks.

**Stage 2:** We incroprate an 2.4B image generation and pretrain for text-to-image generation. We start with a resolution of 256 and progressively increase it to 512 and 1024 during training. **Stage 3:** In the final stage, we jointly train the entire 2.4B + 8B model end-to-end on image understanding, text-to-image generation, image editing, and interleaved generation tasks such as planning and self-reflection. More details on the training data and process are provided in Appendix B.1.

## 4.2 MAIN RESULTS

**Image Understanding.** We report the performance of image understanding tasks in Table 2. Lavida-O outperforms the previous state-of-the-art unified diffusion model, MMaDa, by a considerable margin on MMMU (Yue et al., 2024), MME (Fu et al., 2023), and MMB (Liu et al., 2024c). Compared with the base model LaViDa, Lavida-O achieves substantial improvements on most benchmarks such as ChartQA (Masry et al., 2022), DocVQA (Mathew et al., 2021), ScienceQA (Lu et al., 2022), and MathVista (Lu et al., 2023), due to the scaling of the training data.

Table 2: **Quantitative results on image-level understanding tasks**.*Evaluated by us.

| Model | MMMU | MME-P | MME-C | MMB | ChartQA | DocVQA | InfoVQA | Sci.QA | AI2D | M.Vista | M.Verse |
|---|---|---|---|---|---|---|---|---|---|---|---|
| *AR Und. Only* | | | | | | | | | | | |
| LLaVa-1.6-7B (Liu et al., 2024a) | 35.1 | 1519.3 | 323 | 54.6 | 64.9 | 74.4 | 37.1 | 73.2 | 66.6 | 34.4 | 14.3 |
| Qwen2.5-VL-7B (Bai et al., 2025) | 58.6 | - | - | 83.5 | 84.9 | 82.6 | - | - | 83.9 | 68.2 | 49.2 |
| Intern-VL-3-8B (Li et al., 2024a) | 65.6 | - | - | 83.4 | 86.6 | 92.7 | 76.8 | - | 85.2 | 75.2 | 39.8 |
| *AR Unified Und. and Gen.* | | | | | | | | | | | |
| BAGEL (Deng et al., 2025) | 55.3 | 1687 | 701 | 85 | - | - | - | - | - | 73.1 | - |
| Janus-Pro-1B Chen et al. (2025c) | 36.3 | 1444 | - | 75.5 | - | - | - | - | - | - | - |
| UniGen-1.5B Tian et al. (2025) | 32.3 | - | - | - | - | - | - | 79.4 | 67.4 | 44.6 | - |
| Show-O (Xie et al., 2024) | 27.4 | 1233 | - | - - | - | - | - | - | - | - | - |
| *Masked Und. Only* | | | | | | | | | | | |
| Dimple (Yu et al., 2025b) | 45.2 | 1514 | 432 | 74.6 | 63.4 | - | - | 77.1 | 74.4 | 42.3 | - |
| LaViDa (Li et al., 2025a) | 43.6 | 1366 | 341 | 70.5 | 64.6 | 59.0 | 34.2 | 80.2 | 70.0 | 44.8 | 27.2 |
| *Masked Unified Und. and Gen.* | | | | | | | | | | | |
| Muddit (Shi et al., 2025) | - | 1104 | - | - | - | - | - | - | - | - | - |
| MMaDa (Yang et al., 2025) | 30.2 | 1410 | 242* | 68.5 | 9.8* | 10.9* | 14.9* | 55.8* | 66.6* | 33.7* | 13.5* |
| LaViDa-O | 45.1 | 1431 | 488 | 76.4 | 80.0 | 73.7 | 44.6 | 84.6 | 76.7 | 56.9 | 36.9 |

**Text-to-Image Generation.** We report text-to-image generation results on the GenEval (Ghosh et al., 2023) and DPG (Hu et al., 2024) benchmarks, and FID scores on 30k prompts from the MJHQ (Li et al., 2024b) dataset. We compare against text-to-image models including Flux-dev (Labs, 2024), SD3-Medium (Esser et al., 2024), Meissonic (Bai et al., 2024) and DALLE-3 (OpenAI, 2023), unified models such as BAGEL (Deng et al., 2025), MMaDa (Yang et al., 2025) and Muddit (Shi et al., 2025). Lavida-O significantly outperforms the state-of-the-art Meissonic masked image generation model, as well as unified models such as MMaDa and Muddit. Planning and reflection further enhance prompt-following performance. We did not activate planning and reflection on MJHQ due to its large size and that FID does not reflect prompt-following capabilities.

Table 3: **Quantative results on text-to-image generation tasks.** *Evaluated by us. † For fair comparison, we report results of UniGen after SFT stage.

| Method | Parms. | Type | GenEval ↑ | DPG-Bench↑ | FID-30k↓ |
|---|---|---|---|---|---|
| *Gen. Only* | | | | | |
| Flux-dev (Labs, 2024) | 12B | Continuous | 0.68 | 84.0 | 10.15 |
| SD3-Medium (Esser et al., 2024) | 2B | Continuous | 0.74 | 84.1 | 11.92 |
| DALLE-3 (OpenAI, 2023) | - | Continuous | 0.67 | 83.5 | - |
| Meissonic (Bai et al., 2024) | 1B | Masked | 0.54 | - | - |
| *Unified Und. and Gen.* | | | | | |
| BAGEL (Deng et al., 2025) | 7B+7B | Continuous | 0.82 | - | - |
| Janus-Pro-1B(Chen et al., 2025c) | 1B | AR | 0.73 | 82.6 | - |
| UniGen-1.5B (Tian et al., 2025) | 1B | AR | 0.63† | 82.8 † | - |
| OmniFlow (Li et al., 2024c) | 3.4B | Continuous | 0.62 | - | - |
| Show-o (Xie et al., 2024) | 1.3B | Masked | 0.67 | - | 15.18 |
| Muddit (Shi et al., 2025) | 1B | Masked | 0.61 | - | - |
| MMaDA (Yang et al., 2025) | 8B | Masked | 0.63 | 53.4* | 32.85* |
| LaViDa-O | 4B+2.4B | Masked | 0.77 | 81.8 | 6.68 |
| +Planning | 8B+2.4B | Masked | 0.85 | 82.9 | - |
| +Reflection | 8B+2.4B | Masked | 0.89 | 83.2 | - |

**Object Grounding.** We evaluate the object grounding capabilities of Lavida-O on RefCOCO Referring Expression Comprehension (REC) tasks (Yu et al., 2016; Mao et al., 2016), reporting the Preci-

sion@0.5 metric. Lavida-O outperforms autoregressive vision-language models such as Qwen2.5-VL-7B (Bai et al., 2025) and InternVL3-8B (Zhu et al., 2025), as well as specialist models such as Grounding-DINO-L (Liu et al., 2024b) and SegLLM-7B (Wang et al., 2025a).

Table 4: **Precision@0.5 on RefCOCO, RefCOCO+, and RefCOCOg REC tasks.**

| Model | RefCOCO | | ↑ | RefCOCO+ | | ↑ | RefCOCOg | ↑ |
|---|---|---|---|---|---|---|---|---|
| | val | testA | testB | val | testA | testB | val | test |
| SegLLM-7B(Wang et al., 2025a) | 90.0 | 92.1 | 86.2 | 82.2 | 85.5 | 76.1 | 83.9 | 85.9 |
| Qwen2.5-VL-7B (Bai et al., 2025) | 90.0 | 92.5 | 85.4 | 84.2 | 89.1 | 76.9 | 87.2 | 87.2 |
| GroundingDINO (Liu et al., 2024b) | 90.6 | 93.2 | 88.2 | 88.2 | 89.0 | 75.9 | 86.1 | 87.0 |
| InternVL3-8B (Zhu et al., 2025) | 92.5 | 94.6 | 88.0 | 88.2 | 92.5 | 81.8 | 89.6 | 90.0 |
| LaViDa-O (4-step) | 92.3 | 94.8 | 89.0 | 88.7 | 92.5 | 83.3 | 90.0 | 90.6 |
| LaViDa-O (1-step) | 91.9 | 94.6 | 88.4 | 87.4 | 91.7 | 82.2 | 89.5 | 89.8 |

**Image Editing.** We report image editing results on ImgEdit benchmark (Ye et al., 2025b) in Table 5. Lavida-O outperforms state-of-the-art unified models such as BAGEL and specialized models like FluxKontext-dev. Most notably, Lavida-O even outperforms the state-of-the-art closed-source model GPT4-o(OpenAI, 2024) on replacing and removing objects, which requires localized understanding. This underscores the effectiveness of Lavida-O's design in integrating object-grounding capabilities.

Table 5: **Per-Category and overall scores on ImgEdit benchmark.**

| Model | Add | Adjust | Extract | Replace | Remove | Background | Style | Hybrid | Action | Overall |
|---|---|---|---|---|---|---|---|---|---|---|
| GPT-4o (OpenAI, 2024) | 4.61 | 4.33 | 2.90 | 4.35 | 3.66 | 4.57 | 4.93 | 3.96 | 4.89 | 4.20 |
| Qwen2.5VL+Flux (Wang et al., 2025b) | 4.07 | 3.79 | 2.04 | 4.13 | 3.89 | 3.90 | 4.84 | 3.04 | 4.52 | 3.80 |
| FluxKontext dev (Labs et al., 2025) | 3.76 | 3.45 | 2.15 | 3.98 | 2.94 | 3.78 | 4.38 | 2.96 | 4.26 | 3.52 |
| OmniGen2 (Wu et al., 2025b) | 3.57 | 3.06 | 1.77 | 3.74 | 3.20 | 3.57 | 4.81 | 2.52 | 4.68 | 3.44 |
| UniWorld-V1 (Lin et al., 2025) | 3.82 | 3.64 | 2.27 | 3.47 | 3.24 | 2.99 | 4.21 | 2.96 | 2.74 | 3.26 |
| BAGEL (Deng et al., 2025) | 3.56 | 3.31 | 1.70 | 3.30 | 2.62 | 3.24 | 4.49 | 2.38 | 4.17 | 3.20 |
| Step1X-Edit (Liu et al., 2025) | 3.88 | 3.14 | 1.76 | 3.40 | 2.41 | 3.16 | 4.63 | 2.64 | 2.52 | 3.06 |
| OmniGen (Xiao et al., 2025) | 3.47 | 3.04 | 1.71 | 2.94 | 2.43 | 3.21 | 4.19 | 2.24 | 3.38 | 2.96 |
| UltraEdit (Zhao et al., 2024) | 3.44 | 2.81 | 2.13 | 2.96 | 1.45 | 2.83 | 3.76 | 1.91 | 2.98 | 2.70 |
| AnyEdit (Yu et al., 2025a) | 3.18 | 2.95 | 1.88 | 2.47 | 2.23 | 2.24 | 2.85 | 1.56 | 2.65 | 2.45 |
| InstructAny2Pix(Li et al., 2023b) | 2.55 | 1.83 | 2.10 | 2.54 | 1.17 | 2.01 | 3.51 | 1.42 | 1.98 | 2.12 |
| MagicBrush (Zhang et al., 2023) | 2.84 | 1.58 | 1.51 | 1.97 | 1.58 | 1.75 | 2.38 | 1.62 | 1.22 | 1.90 |
| Instruct-Pix2Pix(Brooks et al., 2023) | 2.45 | 1.83 | 1.44 | 2.01 | 1.50 | 1.44 | 3.55 | 1.20 | 1.46 | 1.88 |
| LaViDa-O | 4.04 | 3.62 | 2.01 | 4.39 | 3.98 | 4.06 | 4.82 | 2.94 | 3.54 | 3.71 |
| + Planning | 4.11 | 3.67 | 2.04 | 4.40 | 4.05 | 4.00 | 4.75 | 3.10 | 4.04 | 3.80 |

## 4.3 TRAINING AND INFERENCE SPEED

In Figure 5, we benchmark the inference efficiency of Lavida-O across three tasks: text-to-image generation, object grounding, and math reasoning. We measure end-to-end latency in seconds per image. Lavida-O is significantly faster than autoregressive models. Notably, we achieve a $6.8\times$ speedup on object grounding tasks compared to Qwen2.5-VL-7B (Bai et al., 2025). We also report the training efficiency measured by per-step latency and compare our Elastic-MoT design with BAGEL-style standard MoT design, Elastic-MoT improves the training speed by $3.17\times$. Specifically, reducing the size of generation branch leads to a speedup of $2.23\times$, and decoupling the attention operation in the last 16 layers lead to an additional speedup of $1.44\times$, We provide additional analysis on the speed-quality tradeoff at inference time in Appendix B.7 and analysis on the training efficiency of Elastic-MoT design in B.2.

## 5 CONCLUSION

In summary, we proposed Lavida-O, the first multi-modal masked diffusion model that achieves state-of-the-art performance on text-to-image generation, image editing, and grounding tasks—competitive with the best specialist models and autoregressive unified models. We also introduced a novel paradigm of interleaved generation, which explicitly leverages understanding capabilities to improve generation results in a unified multi-modal model through planning and self-reflection. In developing Lavida-O, we proposed several efficient training and inference techniques, including the ElasticMoT architecture, universal text conditioning, and stratified random sampling, providing valuable insights for future work in masked diffusion models and unified multi-modal systems.

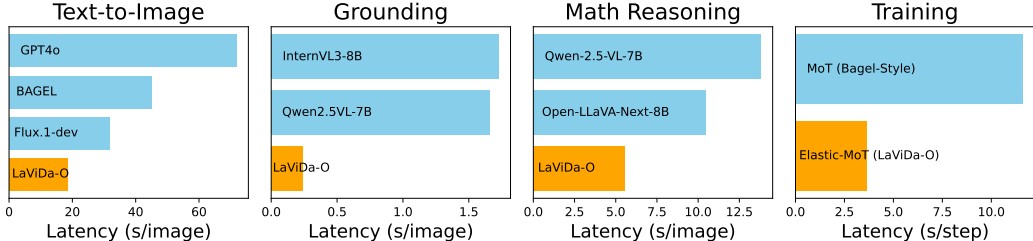

Figure 5: **Training and Inference Speed of Lavida-O.** We compare the end-to-end inference latency of Lavida-O on three tasks, as well as pretraining efficiency measured by per-step latency.

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

# A ADDITIONAL TECHNICAL DETAILS

## A.1 FORMULATION OF MASKED DIFFUSION MODELS

Masked Diffusion Models (MDMs) model the generation process of discrete token sequences through a continuous-time Markov chain (CMTC). Formally, given a sequence of discrete tokens $X_0 = [X_0^1, X_0^2, \ldots, X_0^L]$ of length $L$, the forward process $q(X_t|X_s)$ gradually converts it into a sequence of mask tokens $[M]$, denoted by $X_1 = [X_1^1, X_1^2, \ldots, X_1^L]$, over the continuous time interval $[0, 1]$, with $1 \geq t \geq s \geq 0$. Each token $X_t^i$ belongs to a fixed-size vocabulary set $V$. In our setup, $V$ consists of text tokens, image VQ tokens, and the special mask token $[M]$. This forward process is formally defined as

$$q(X_t^i|X_s^i) = \begin{cases} \text{Cat}(X_t^i; \mathbf{M}), & \text{if } X_s^i = [M] \\ \text{Cat}(X_t^i; \frac{1-t}{1-s}\mathbf{X_s^i} + \frac{t-s}{1-s}\mathbf{M}), & \text{if } X_s^i \neq [M], \end{cases} \tag{2}$$

where $\text{Cat}(\cdot)$ denotes a categorical distribution, and $\mathbf{M}, \mathbf{X_0^i}, \mathbf{X_s^i} \in \mathbb{R}^{|V|}$ are probability vectors, with $|V|$ denoting the vocabulary size. In particular, $\mathbf{M}$ is a one-hot vector representing the mask token $[M]$. This forward process yields the following marginal distribution:

$$q(X_t^i|X_0^i) = \text{Cat}(X_t^i; (1-t)\mathbf{X_0^i} + t\mathbf{M}). \tag{3}$$

MDLM (Sahoo et al., 2024) demonstrated that the posterior of the reverse process $p(X_s|X_t, X_0)$ has the following form:

$$p(X_s^i|X_t^i, X_0^i) = \begin{cases} \text{Cat}(X_s^i; \mathbf{X_t^i}), & \text{if } X_s^i \neq [M] \\ \text{Cat}(X_s^i; \frac{t-s}{t}\mathbf{X_0^i} + \frac{s}{t}\mathbf{M}), & \text{if } X_s^i = [M]. \end{cases} \tag{4}$$

In practice, we replace $\mathbf{X_0^i}$ with the neural network prediction $p_\theta(X_0^i|X_t)$ when sampling from the reverse process, which gives the following transition:

$$p_\theta(X_s^i|X_t) = \begin{cases} \text{Cat}(X_s^i; \mathbf{X_t^i}), & \text{if } X_s^i \neq [M] \\ \text{Cat}(X_s^i; \frac{t-s}{t}p_\theta(X_0^i|X_t) + \frac{s}{t}\mathbf{M}), & \text{if } X_s^i = [M]. \end{cases} \tag{5}$$

**Sampling process.** At inference time, we initialize $X_1$ as a sequence of mask tokens, with $X_1^1 = X_1^2 = \cdots = X_1^L = [M]$. We discretize the continuous time interval $[0, 1]$ into discrete timesteps $0 = t_0 < t_1 < \cdots < t_K = 1$, and iteratively sample $X_{t_{k-1}} \sim p_\theta(X_{t_{k-1}}|X_{t_k})$ using Equation 5. We start with $k = K$ and end when we obtain a mask-free sequence $X_0$. At each step, we sample each token position independently, assuming that $p_\theta(X_{t_{k-1}}|X_{t_k})$ factorizes as $\prod_{i=1}^L p_\theta(X_{t_{k-1}}^i|X_{t_k})$, following previous works (Nie et al., 2025; Sahoo et al., 2024; Lou et al., 2023).

**Training process.** At each training step, given a clean sequence $X_0$, we sample a random timestep $t \in [0, 1]$ and obtain $X_t \sim q(X_t|X_0)$ through the forward process defined in Equation 3. The loss is then computed using Equation 1 from Section 2.1.

In this section, we have documented the standard training and inference process for typical MDMs. Our modality-aware masking design introduces several modifications to the above processes, which are described in Section 3.1.2 of the main paper. Additional details are provided in Appendix A.3.

## A.2 ELASTIC-MOT ARCHITECTURE

In this section, we document the detailed design of the Elastic-MoT architecture described in Section 3.1.1. As discussed in the main paper, the proposed Elastic-MoT architecture has two key differences compared to standard MoT: a generation branch with variable size and decoupled joint attention in the later layers.

**Variable-sized generation branch.** In standard MoT models such as BAGEL (Deng et al., 2025), the generation branch is initialized as an exact copy of the understanding branch. For models in the

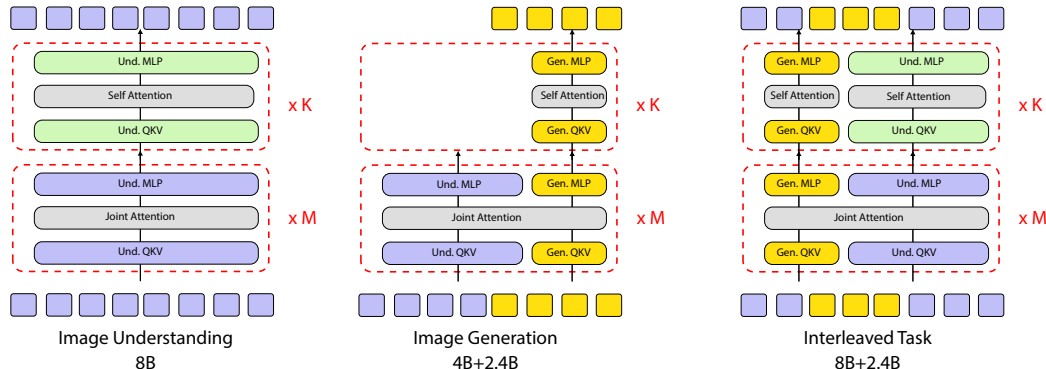

Figure 6: **Activated parameters of Lavida-O under different task settings.** Elastic-MoT Design allow Lavida-O to dynamically loads its parameters depending on the tasks. For understanding-only tasks, we only load the 8B generation branch. For text-to-image generation tasks, we load the first $M = 16$ layers of the understanding branch, which consists of 4B parameters, and the full 2.4B generation branch. For interleaved tasks, we load all 2.4B+8B parameters.

7–10B scale, this leads to a substantial increase in parameter count and compute overhead, limiting the scalability of MoT models. Motivated by the success of many medium-sized, high-quality text-to-image generation models, we explore using a smaller generation branch in the Elastic-MoT design. Since we still want the modalities to interact with each other through the joint attention mechanism, it is important to keep the dimensions of the query and the key vectors consistent. We provide a detailed breakdown of the parameter sizes in Table 6. To initialize the generation branch with dimensions smaller than the understanding branch, we truncate the weights of the understanding branch and copy them to the generation branch.

Table 6: **Comparison of understanding (Und) branch and generation (Gen) branch configurations.** The projection sizes are in the format [output_size, input_size].

|  | Und Branch | Gen Branch |
|---|---|---|
| | *Attention* | |
| norm | 4096 | 2048 |
| q_proj_size | [4096, 4096] | [4096, 2048] |
| k_proj_size | [4096, 4096] | [4096, 2048] |
| v_proj_size | [4096, 4096] | [4096, 2048] |
| attn_out | [4096, 4096] | [4096, 2048] |
| | *MLP* | |
| norm | 4096 | 2048 |
| input_size | 4096 | 2048 |
| hidden_size | 12288 | 8192 |
| output_size | 4096 | 2048 |

**Decoupled attention.** In standard MoT, understanding and generation tokens can interact with each other in all $N$ transformer layers through the joint attention mechanism. We decouple attention in the last $K$ layers and only allow tokens of the same type to interact with each other. In the first $M = N - K$ layers, all tokens can still interact with each other as in the standard MoT architecture. This design is motivated by two factors. First, it prevents text and image tokens from interfering with each other's representations in the later stages of generation. Second, and more importantly, it allows us to load only 4B out of 8B parameters for text-to-image generation tasks, greatly improving the scalability of pretraining while also reducing compute cost at inference time. We visualize the activated parameters for different tasks in Figure 6. For understanding-only tasks, we activate only the understanding branch in all $N = M + K$ layers. For generation-only tasks, we activate the understanding branch in the first $M$ layers and the generation branch in all $N = M + K$ layers.

For interleaved tasks with both text and image outputs, we activate all parameters. In our setup, we choose $M = K = 16$, which yields $N = 32$ layers in total.

### A.3 MODALITY-AWARE MASKING

In this section, we provide details of the changes to the training and sampling process introduced by modality-aware masking, as described in Section 3.1.2. Recall that in adapting the MoT architecture for MDMs, one of the main challenges is routing tokens. In particular, while we can easily decide which branch should process unmasked tokens based on whether they are image VQ tokens or text tokens, it is difficult to make such decisions for masked tokens, especially in interleaved generation tasks where the final output contains both images and text. Modality-aware masking addresses this problem by processing all tokens with the understanding branch by default and dynamically deciding when and where to invoke the generation branch during the sampling process.

**Sampling Process.** For convenience, we denote masked tokens that will be processed by the understanding branch as $M_{und}$ and masked tokens that will be processed by the generation branch as $M_{gen}$. With this distinction, the routing policy becomes simple: all text tokens plus $M_{und}$ are processed by the understanding branch, while all image VQ tokens plus $M_{gen}$ are processed by the generation branch. We introduce a special text token [exp] to indicate when an image should be generated. When a [exp] token is generated in the unmasking process, it is automatically replaced with a sequence of $M_{gen}$ tokens. The number of $M_{gen}$ tokens representing each image is determined by prespecified output resolution. These tokens are then processed by the generation branch in subsequent rounds. For example, each $1024 \times 1024$ image is represented by 1024 VQ tokens. This process is documented in Algorithm 1 and illustrated in Figure 7 (Left).

---

**Algorithm 1** Interleaved Generation with Modality-Aware Masking

---

**Input:** Initial Generation Length $L$, discrete timestamps $0 = t_0 < t_1 < \cdots < t_K = 1$, prompt $C$
1: Initialize $t \leftarrow K$
2: Initialize $X_t^{1:L} \leftarrow M_{und}$
3: **for** $i = T$ to 1 **do**
4:      Sample $X_{t_{i-1}} \sim p_\theta(X_{t_{i-1}} \mid X_{t_k}, C)$    *// Eq. 5*
5:      **if** a [exp] token is generated in $X_{t_{i-1}}$ **then**
6:          Replace it with a sequence of $M_{gen}$ tokens
7:          *// These $M_{gen}$ will be routed to the generation branch in subsequent rounds*
8:      **end if**
9: **end for**
10: **return** Fully unmasked sequence $X_0$

---

**Training.** A consequence of modality-aware masking is that the partially masked sequence $X_t$ will have varying length depending on $t$, making the loss described in Equation 1 not directly applicable. In particular, when sampling from the forward process $q(X_t \mid X_0)$, there is a special timestep $t_{exp}$ at which a sequence of VQ image tokens is collapsed into a single [exp] text token. As illustrated in Figure 7 (Right), when $t < t_{exp}$, $X_t$ has a shorter sequence length than $X_0$. To apply the loss properly, we construct a new sequence $X_0'$ by collapsing all sequences of image VQ tokens into [exp] tokens in $X_0$. We then modify the loss in Equation 1 to the following:

$$\mathcal{L}_{\text{MDM}} = -\mathbb{E}_{t,X_0,X_t} \left[ \frac{1}{t} \sum_{\{i \mid X_t^i = [M]\}} \log p_\theta(\hat{X}_0^i \mid X_t) \right], \tag{6}$$

$$\text{where } \hat{X}_0 = \begin{cases} X_0, & \text{if } t \in (t_{\text{exp}}, 1) \\ X_0', & \text{if } t \in (0, t_{\text{exp}}) \end{cases} \tag{7}$$

This change is also highlighted in blue in Figure 7 (Right).

**Understanding-Only and Generation-Only Tasks.** We activate modality-aware masking only for interleaved tasks, since these require both the understanding and generation branches in our Elastic-MoT architecture. For computational efficiency, we do not use modality-aware masking for

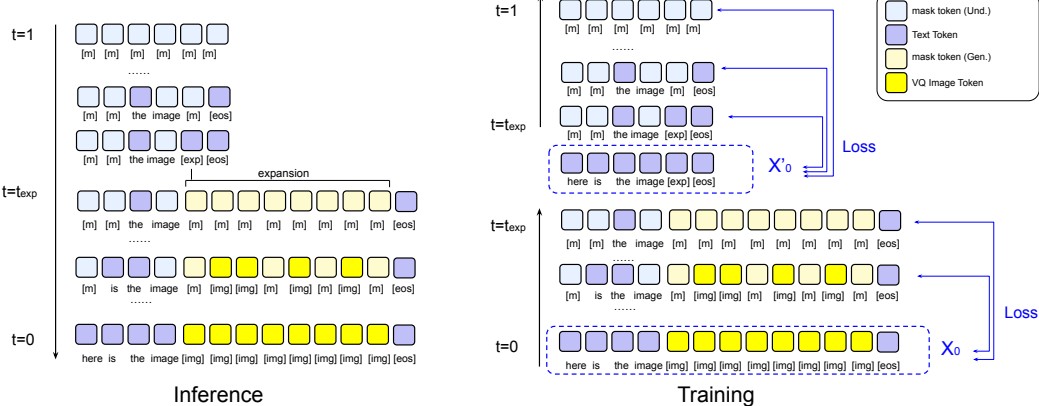

Figure 7: **Training and inference with modality-aware masking.** We visualize the sampling process with modality-aware masking on the left and the training process on the right. During the training, the loss is applied on either $X_0$ or $X'_0$ depending on the value of $t$ with respect to $t_{\text{exp}}$.

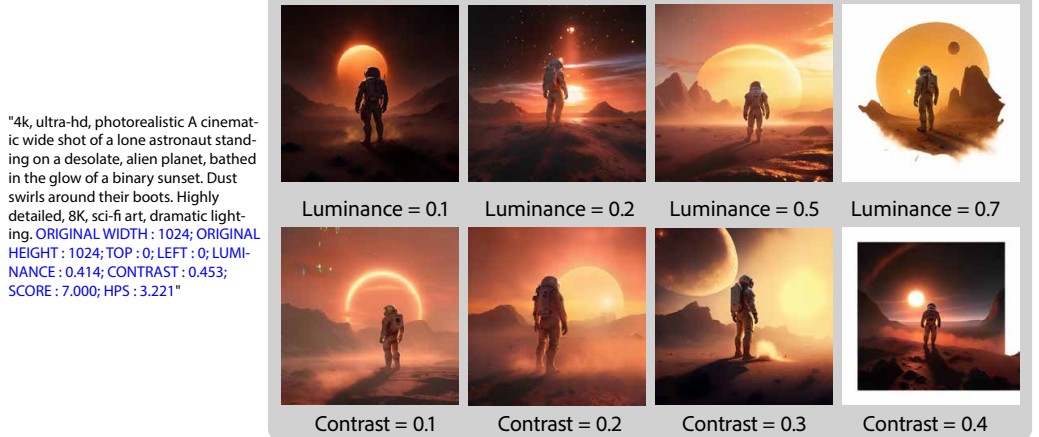

Figure 8: **Effect of Universal Text Conditioning.** On the left side, we visualize the text format used in Universal Text Conditioning. On the right side, we visualize generation results under different choices of universal text conditioning.

understanding-only tasks such as image captioning, or for generation-only tasks such as text-to-image generation (without planning and reflection). This allows us to best utilize the flexibility of Elastic-MoT and avoid loading unnecessary model parameters.

## A.4 UNIVERSAL TEXT CONDITIONING

Universal text conditioning is inspired by the micro-conditioning approach (Podell et al., 2023) employed in many text-to-image models. These models interoperate special conditioning embeddings to incorporate non-text conditions such as the original image resolution or aesthetic score. Since Lavida-O is a unified model with mathematical reasoning capabilities, we can represent these conditions directly as plain text. In particular, we include source image resolution, crop coordinates, aesthetic scores (Schuhmann, 2022), and HPS scores (Wu et al., 2023), following existing works (Podell et al., 2023; Bai et al., 2024). Additionally, we incorporate luminance (brightness) and contrast to give users greater control over the generated images. Each condition is represented as a simple string of the form "[KEY] : [VALUE]". During training, each condition is randomly dropped with some probability. At inference, users may specify all conditions or only a subset.

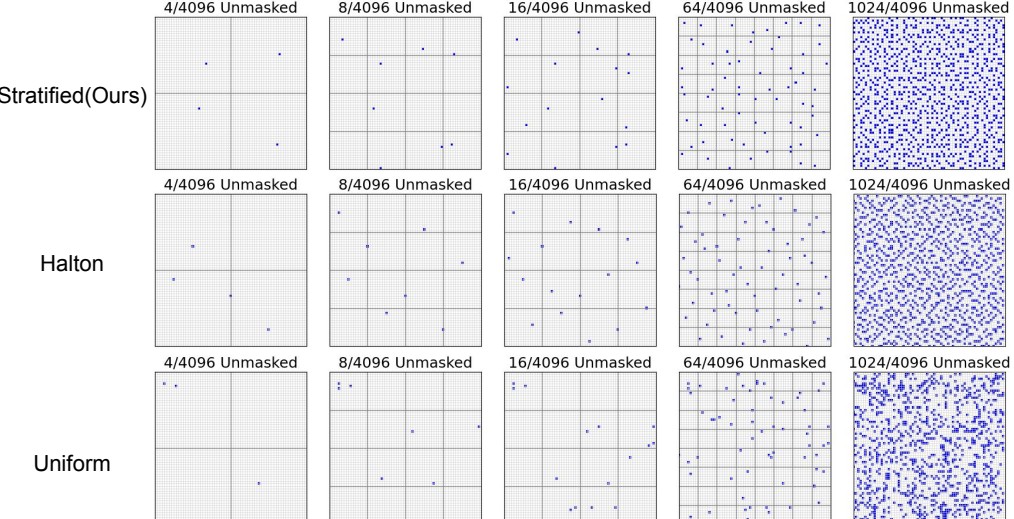

Figure 9: **Visualization of different sampling processes.** We compare the unmasking order of the stratified sampler, Halton sampler, and uniform random sampler. Uniform random sampler produces the least desirable spatial pattern, with many unmasked tokens clustered together. Halton sampler is less ideal than stratified sampler because it does not guarantee perfectly stratified coverage. For example, when the number of unmasked tokens is 4, the upper-right quadrant remains unoccupied.

This design is illustrated in Figure 8. By modifying these universal text conditioning parameters at inference time, users can flexibly control various image properties such as brightness. Notably, when brightness and contrast are set to very high values, the generated images become highly stylized in order to satisfy the constraints.

## A.5    Stratified Random Sampling

In this section, we provide detailed descriptions of the stratified random sampling process introduced in Section 3.2. In the vanilla sampling process described in Equation 5, each token is unmasked independently. In practice, this often leads to suboptimal generation quality. Instead of unmasking tokens randomly, several works adopt alternative sampling strategies in which the unmasking order of tokens is determined by heuristics such as the model's confidence at each token position (Nie et al., 2025; Ye et al., 2025a; Chang et al., 2022).

In image generation, tokens with high confidence are frequently adjacent to one another. As a result, confidence-based unmasking tends to reveal many adjacent tokens in a single step. Since tokens that are spatially adjacent are often highly correlated, this violates the independence assumption $p_\theta(X_{t_{k-1}}|X_{t_k}) = \prod_{i=1}^{L} p_\theta(X_{t_{k-1}}^i|X_{t_k})$ stated in Section A.1. To address this, we design a stratified sampling process that ensures the unmasked tokens are spatially dispersed. Specifically, we enforce that the first 4 unmasked tokens occupy the four quadrants of the image; the first 16 unmasked tokens occupy all 16 subregions obtained by dividing the image into a $4 \times 4$ grid; and so forth. The algorithm is formally described below:

Our design is inspired by the stratified sampling process commonly used in numerical integration and computer graphics. It also follows a similar motivation to the recent Halton mask scheduler, which uses the low-discrepancy Halton sequence to ensure that unmasked tokens are spatially dispersed (Besnier et al., 2025). We illustrate the differences among stratified sampling, Halton sampling, and uniform random sampling in Figure 9. As shown in the figure, uniform random sampling produces the least desirable spatial pattern, with many unmasked tokens clustered together. Compared with our proposed stratified sampling process, Halton sampling is less ideal because it does not guarantee perfectly stratified coverage. For example, when the number of unmasked tokens is 4, the upper-right quadrant remains unoccupied. The benefits of stratified sampling are also reflected in FID scores, which we document in Section B.3.

---

**Algorithm 2** Stratified Unmasking Order

---

**Input:** Image size $N \times N$
**Output:** a list $\mathcal{O}$ of coordinates $(i, j)$ indicating unmasking order
 1: Initialize an empty list $\mathcal{O}$
 2: **for** $d = 1, 2, \ldots, \log_2 N$ **do**
 3:     Partition the image into $2^d \times 2^d$ grid cells
 4:     **for** each grid cell $g$ in random order **do**
 5:         **if** $\mathcal{O} \cap g = \varnothing$ **then**
 6:            Sample $(i_g, j_g)$ uniformly within cell $g$
 7:            Append $(i_g, j_g)$ to $\mathcal{O}$
 8:         **end if**
 9:     **end for**
10: **end for**
11: **return** $\mathcal{O}$

---

### A.6 OBJECT GROUNDING WITH COORDINATE QUANTIZATION

In this section, we provide detailed descriptions of Lavida-O's design for object grounding tasks. Given an image and a referring expression describing an object, the grounding task requires locating the described object in the image by predicting its bounding box coordinates. In autoregressive vision-language models such as Qwen2.5-VL (Bai et al., 2025), bounding boxes are represented as plain text strings, such as "[123, 232, 300, 1021]". At inference, the coordinates are generated sequentially from left to right. This design has several limitations. First, since the model only sees a padded and resized image, it is difficult for the model to predict absolute pixel coordinates that depend on the original resolution of the input image. Second, the sequential generation order is slow and inefficient.

To address these issues, we normalize the bounding box coordinates and quantize them into discrete bins. Specifically, given an image of size $H \times W$, we first pad it to a square image of size $D \times D$, where $D = \max(H, W)$, and normalize the bounding boxes in the padded image to the range [0,1] by dividing the raw pixel coordinates by $D$. This step makes the coordinates independent of the original input resolution. We then round each coordinate into 1025 bins representing $\frac{0}{1024}, \frac{1}{1024}, \frac{2}{1024}, ..., \frac{1024}{1024}$ and represent them with special tokens. This reduces the number of tokens needed to represent each bounding box to exactly 4. Finally, since Lavida-O is a masked diffusion model with a bi-directional attention mask and parallel decoding capabilities, we can predict multiple bounding boxes simultaneously. For example, if we want to obtain the bounding boxes of both "a cute dog" and "a boy," we can initialize a text sequence "a cute dog [m][m][m][m]; a boy [m][m][m][m]" and perform parallel unmasking of multiple bounding box coordinates. This design is illustrated in Figure 10.

### A.7 REFLECTION AND PLANNING

The unique advantage of unified understanding and generation models is that they can leverage their understanding capabilities to improve generation results. Several works on unified models show that simple joint training on a combination of understanding and generation tasks improves performance on generation tasks (Xie et al., 2024; Deng et al., 2025), particularly in instruction-following capabilities. Lavida-O pushes this paradigm further by introducing two explicit mechanisms to exploit understanding capabilities: planning and reflection. At inference, these capabilities are invoked via specialized prompts, such as "please generate a layout design before creating the final image".

**Planning.** To improve prompt-following capabilities in text-to-image generation, we ask the model to first generate a layout design of objects, which consists of (object, bounding box) pairs, before generating the final image. Such interleaved generation is achieved through the modality-aware masking process described in Section A.3. We illustrate this process in Figure 11 (Top). As shown, planning enables Lavida-O to follow challenging and unintuitive prompts, such as "a horse *above* an astronaut."

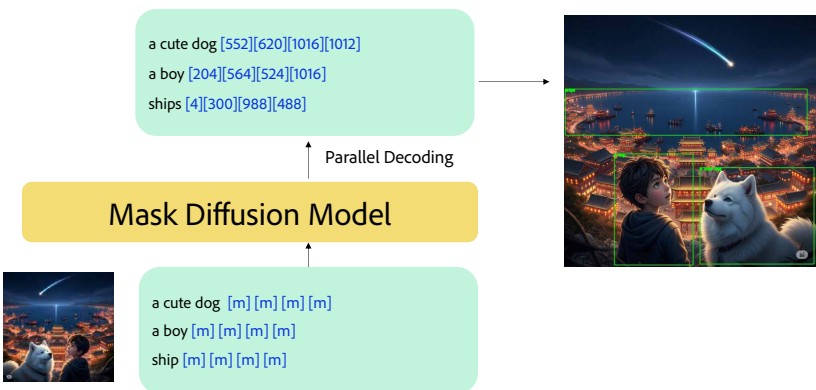

Figure 10: **Coordinate Quantization**. We normalize bounding box coordinates into the range [0,1] and discretize them into 1025 bins. This ensures that each bounding box is represented by exactly 4 tokens, allowing efficient parallel decoding of multiple bounding boxes in a single step.

Similarly, we can adopt planning for image editing tasks. Given an input image and an edit instruction, the model can first leverage its grounding capabilities to identify the regions that need to be edited before generating the edited image. This process is illustrated in Figure 11 (Bottom).

**Reflection.** We can improve text-to-image generation performance by leveraging Lavida-O's understanding capability to achieve self-critique and iterative self-improvement. Given an input prompt, the model first generates an image, then performs a self-critique step to evaluate whether the generated image matches the prompt. If it does, the generation process terminates. Otherwise, the model generates a revised image and attempts to fix the identified issues. This cycle is repeated until an image passes the self-critique process or the maximum number of rounds is reached. At each round, we also invoke the planning capability. Since Lavida-O's context length is limited to 8192 tokens, we truncate the history when necessary to include at most three rounds. This process is illustrated in Figure 11 (Middle). Formally, the reflection process is defined by the following algorithm

---

**Algorithm 3** Iterative Image Refinement with Self-Reflection Loop

---

**Input:** Text prompt $P$, Unified Model $\Theta$, Max Iterations $N$
**Output:** Output image $I$
1: Initialize $I_1 \leftarrow$ GenerateWithPlanning$(\Theta, P)$                ▷ Generate an initial image
2: $F_1 \leftarrow$ GetTextFeedback$(\Theta, P, I_1)$                ▷ Obtain initial feedback
3: **for** $i = 2$ to $N$ **do**
4:     $\mathcal{H}_i \leftarrow \{(I_j, F_j) \mid j = 1, 2, ..., i - 1\}$                ▷ Construct History Context
5:     **if** Run out of context limit of model $\Theta$ **then**
6:         Truncate $\mathcal{H}_i$ by removing early rounds
7:     **end if**
8:     $I_i \leftarrow$ GenerateWithPlanning$(\Theta, P, \mathcal{H}_i)$                ▷ Generate a new image
9:     $F_i \leftarrow$ GetTextFeedback$(P, I_i)$                ▷ Obtain new feedback
10:     **if** $F_i = $ `""` **then**                ▷ Stop if no more improvements
11:         **return** $I_i$
12:     **end if**
13: **end for**
14: **return** $I_N$

---

Similar designs and algorithms have been explored for generation-only models in the context of inference-time scaling, such as Reflect-DiT (Li et al., 2025b) and ReflectionFlow (Zhuo et al., 2025). However, unlike these works, which require an external vision-language model as a reward model, Lavida-O uniquely unifies layout planning, self-critique, and iterative self-improvement in a single model through a unified generation process.

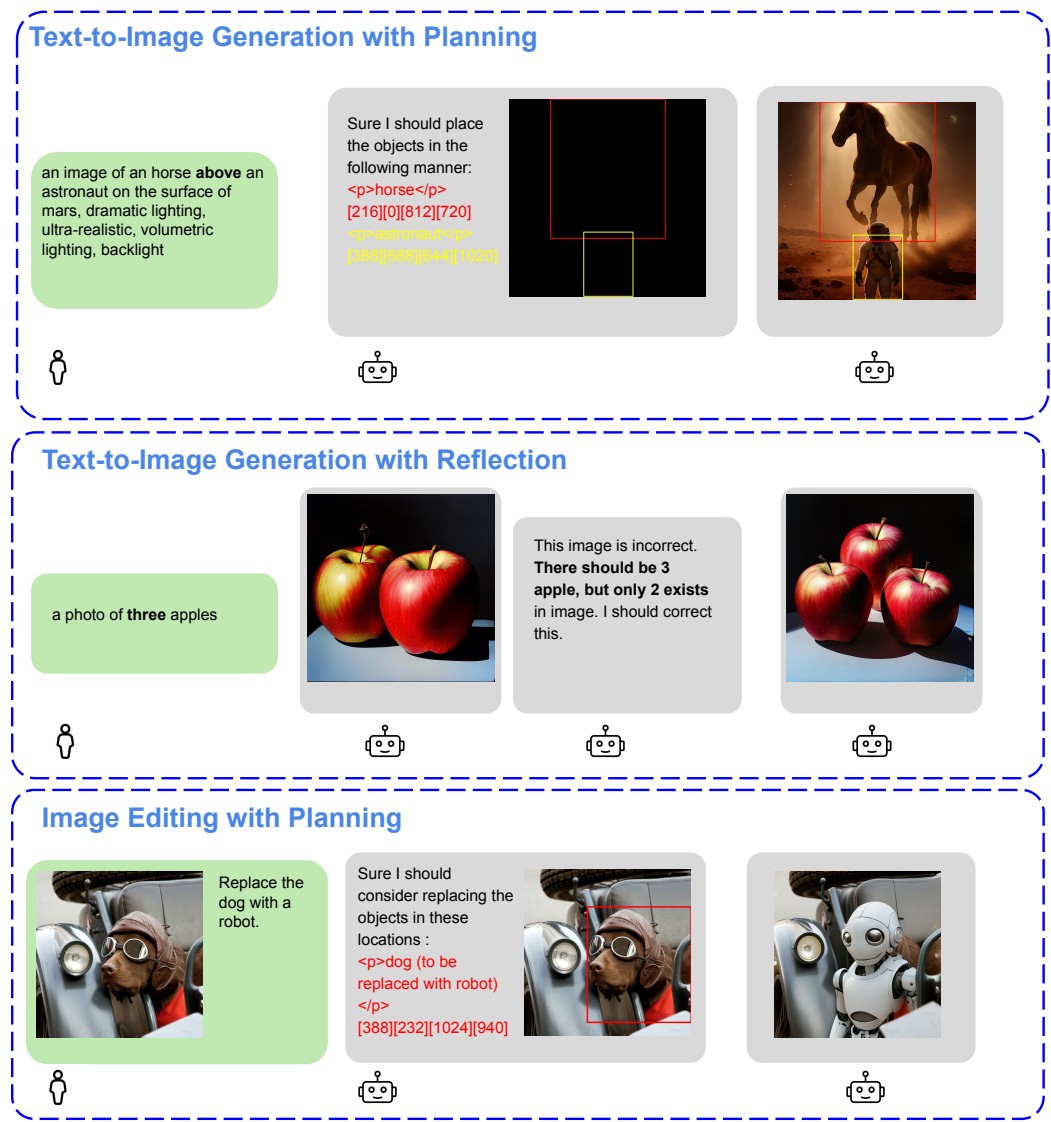

Figure 11: **Interleaved Generation with Planning and Reflection.** We provide visual examples of interleaved generation, including text-to-image generation with planning (Top), text-to-image generation with reflection (Middle), and image editing with planning (Bottom). We always enable planning during the reflection process. The layout traces is omitted in the middle figure for clarity and better presentation.

## B  ADDITIONAL EXPERIMENT DETAILS AND RESULTS

In this section, we document the details of the experiments for better reproducibility, including data pipeline, training hyperparameters, and compute cost. In addition, we also provide additional experimental results on the effectiveness of various design choices used by Lavida-O, such as the Elastic-MoT design, stratified sampling, and the data pipeline.

### B.1  SETUP

**Pretrained Weights.** We use LaViDa (Li et al., 2025a) to initialize the understanding branch and semantic encoder. For the VQ encoder, we adopt the Meissonic encoder (Bai et al., 2024). The

image generation branch is initialized from the truncated weights of the understanding branch, as described in Section A.2.

**Data Pipeline.** Unlike many frontier models, our model does not make use of proprietary images or documents. Our training data consists of the following components:

- *A: Text-to-Image Pairs.* We source data from LAION-2B (Schuhmann et al., 2022) and COYO-700M (Byeon et al., 2022). We additionally include SA-1B (Kirillov et al., 2023), JourneyDB (Sun et al., 2023), BLIP3o-60k (Chen et al., 2025a), and ShareGPT4o-Image (Chen et al., 2025b). Each dataset is heavily filtered to remove NSFW prompts, low CLIP scores (Radford et al., 2021), low aesthetic scores (Schuhmann, 2022), and low-resolution images. This results in 200M images in our final mix. Where available, we use captions generated by VLMs instead of raw alt-texts. These captions are sourced from existing work including Recap-LAION, Recap-COYO (Wu et al., 2024), and BLIP-3o (Chen et al., 2025a).

- *B: Image-level Understanding Data.* We include LLaVA-OneVision (Li et al., 2024a), Open-LLaVA-Next (Chen & Xing, 2024), MAmmoth-VL (Guo et al., 2024), and Visual-WebInstruct (Jia et al., 2025).

- *C: Region-level Understanding Data.* We include GranD (Rasheed et al., 2024) and Ref-COCO (Kazemzadeh et al., 2014).

- *D: Image Editing Data.* We include ShareGPT4o-Image (Chen et al., 2025b), GPT-Edit-1.5M (Wang et al., 2025b), and the image editing subset of UniWorld-V1 (Hu et al., 2022).

- *E: Interleaved Planning and Reflection Data.* For planning data, we manually construct a layout dataset by running an open-vocabulary object detector, GroundingDino-L (Liu et al., 2024b), on the outputs of image generation and editing datasets, including BLIP-3o (Chen et al., 2025a), ShareGPT4o-Image (Chen et al., 2025b), and GPT-Edit-1.5M (Wang et al., 2025b). For reflection data, we leverage existing datasets including ReflectDiT (Li et al., 2025b) and ReflectionFlow (Zhuo et al., 2025).

**Training Setup.** Training consists of three stages. In the first stage, we extend LaViDa to region-level tasks such as grounding. In the second stage, we perform large-scale pretraining on text-to-image generation tasks. In the final stage, we jointly train the model on a mix of understanding, generation, and interleaved tasks. We document the training hyperparameters, the datasets used, the active parameter count, and other relevant details in Table 7.

In addition, we implement a dataset mix scheduler that dynamically adjusts the sampling weight of each dataset throughout training to address data imbalance. Specifically, we assign a high weight to new capabilities at the beginning of each training stage and gradually decay the weight over time. For example, in Stage 1 we have fewer than 1M grounding samples but more than 10M image-level understanding samples. To enable efficient acquisition of grounding capability while preventing overfitting, we initially set the grounding-to-understanding ratio to 3:1, which is gradually decreased to 1:3. We provide further analysis of the scheduler in Section B.4.

## B.2 Ablation Studies on Elastic-MoT Design

In this section, we report ablation results of the Elastic-MoT design, including the size of the generation branch and the number of joint attention layers.

**Size of Generation Branch.** We report the performance of Lavida-O with different sizes of the generation branch during text-to-image pretraining (Stage 2) in Table 8. The results are obtained after 50k training steps with a global batch size of 1024. We also document the maximum per-GPU batch size, the gradient accumulation steps, and the training latency to measure efficiency. The results show that models of different sizes achieve comparable performance after 50k steps. Smaller models (1B, 2B) converge slightly faster and achieve marginally higher performance than larger models (4B, 8B). Larger models (4B, 8B) are harder to optimize as they need more steps, data, and tuning to realize their full capacity. In terms of latency, smaller models are considerably faster. The 2B model achieves the best balance between performance and efficiency, attaining the highest GenEval and DPG scores while being $3.17\times$ faster.

Table 7: **Training configurations across three stages.** We use letters A-E to represent different dataset following Section B.1.

|  | Stage 1 | Stage 2 | Stage 3 |
|---|---|---|---|
| Learning Rate | $5 \times 10^{-6}$ | $1 \times 10^{-4}$ | $2 \times 10^{-5}$ |
| Steps | 80k | 400k | 100k |
| $\beta_1$ | 0.99 | 0.99 | 0.99 |
| $\beta_2$ | 0.999 | 0.999 | 0.999 |
| optimizer | AdamW | AdamW | AdamW |
| Dataset Used | B,C | A | A,B,C,D,E |
| Loaded Parameters | 8B | 6.4B | 10.4B |
| Trainable Parameters | 8B | 2.4B | 10.4B |
| Und. resolution | $384 \times \{(1,3),(2,2)\}$ | $384 \times \{(1,3),(2,2)\}$ | $384 \times \{(1,3),(2,2)\}$ |
| Gen. resolution | - | $256 \to 512 \to 1024$ | 1024 |
| Semantic Encoder | Trainable | Not Loaded | Trainable |
| VQ Encoder | Not Loaded | Loaded | Loaded |
| Gen. Branch | Not Loaded | Trainable | Trainable |
| Und. Branch | Trainable | Partially Loaded | Trainable |

Table 8: **Comparison of different model sizes on GenEval, DPG, and training efficiency.** We report the performance of Lavida-O with different sizes of the generation branch during the text-to-image pretraining (Stage-2) after 50k training steps. We also report the per-GPU batch size and training latency.

| Architecture | | Performance | | Efficiency | | |
|---|---|---|---|---|---|---|
| Parm. | Hidden Size | GenEval ↑ | DPG ↑ | Batch Size | Accum. Step | Latency (s/it)↓ |
| 4B+1B | 1536 | 0.56 | 60.8 | 16 | 1 | **1.98** |
| 4B+2B | 2048 | **0.57** | **63.1** | 16 | 1 | 3.67 |
| 4B+4B | 3072 | 0.48 | 55.3 | 8 | 2 | 8.42 |
| 4B+8B | 4096 | 0.55 | 58.6 | 8 | 2 | 11.64 |

**Number of Joint Attention Layers.** To study the effect of varying the number of joint attention layers, we conducted two ablation experiments. The first experiment was performed during Stage 2 pretraining. We started with the Stage 1 checkpoint with $N = 32$ layers in the understanding branch and fixed the generation branch size to 4B. We then varied $M$, the number of layers with joint attention, among $\{8,16,24,32\}$. The number of non-joint layers, $K$, is automatically determined by $K = N - M$. The results after 100k training steps are shown in Table 9. Among the four choices, $M = \{16, 24, 32\}$ yields a comparable performance, while $M = 8$ shows a substantial drop. This suggests that a sufficient number of joint attention layers is necessary for strong text-to-image performance, but additional layers beyond a threshold provide little benefit. Training latency also decreases when $M$ is smaller (i.e., larger $K$), as fewer joint layers must be loaded. $M = 16$ achieves the best balance of speed and performance.

Table 9: **Effect of varying choices of M and K in partially-decoupled attention design.** Efficiency is measured in Stage-2 training. For stage-3 training, we need to load all layers since the data contain a mix of text, image, and interleaved generation tasks.

| M | K | Pretraining (Stage-2) | | SFT (Stage-3) | | | Efficiency |
|---|---|---|---|---|---|---|---|
|  |  | GenEval ↑ | DPG ↑ | GenEval ↑ | DPG↑ | ImageEdit↑ | Latency (s/it)↓ |
| 8 | 24 | 0.57 | 69.3 | - | - | - | **2.45** |
| 16 | 16 | **0.63** | **75.0** | **0.89** | 83.2 | **3.66** | 3.67 |
| 24 | 8 | 0.63 | 73.3 | 0.81 | 83.0 | 3.60 | 4.12 |
| 32 | 0 | 0.61 | 71.2 | 0.85 | **83.2** | 3.55 | 5.20 |

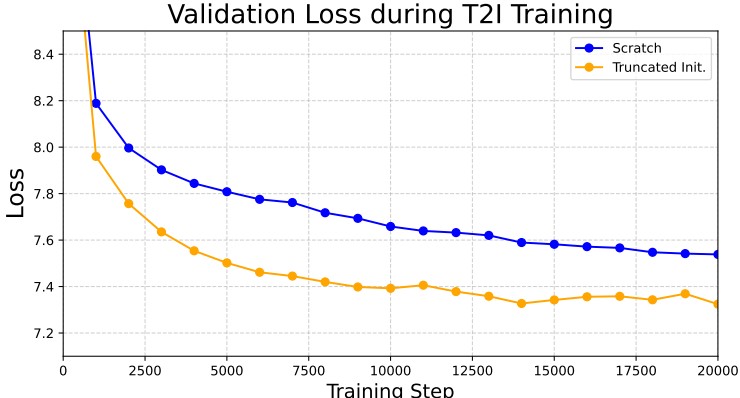

Figure 12: **Effect of truncated initialization.** Validation loss comparison of truncated initialization vs. training from scratch during Stage 2. Truncated initialization converges faster and achieves lower loss.

We conducted a second experiment in Stage 3, where interleaved generation and editing tasks may benefit more from joint attention. Starting from a Stage 2 checkpoint pretrained with $M = 16$ layers for 400k steps, we trained for 50k steps under $M = \{16, 24, 32\}$. The results show two key observations: (1) text-to-image tasks converge faster than image-editing tasks, reaching near-final performance after 50k steps, while editing tasks lag behind; (2) increasing $M$ does not significantly improve performance, even for interleaved editing. This may be due to token interference at later layers or the Stage 2 model being optimized with only 16 joint layers. Due to compute constraints, we were unable to retrain Stage 2 with alternative values of $M$. Nevertheless, keeping $M = 16$ is a reasonable choice given our setup. Finally, in Stage 3 the efficiency difference is less pronounced, since all 10.4B parameters must be loaded for interleaved training and inference.

**Weight Initialization.** We initialized the 2.4B generation branch with truncated weights from the understanding branch (Section A.2). We also explored initializing from scratch. Figure 12 shows the validation loss during the first 20k steps of Stage 2. Truncated initialization converges faster and yields lower loss.

### B.3 ABLATION STUDIES ON STRATIFIED SAMPLING

We compared image generation quality under different sampling strategies on the MJHQ-30k dataset (Li et al., 2024b) with 64 sampling steps. We evaluate the proposed stratified sampler against confidence-based sampling (Chang et al., 2022), uniform random sampling, and Halton sampling (Besnier et al., 2025). The results are reported in Table 10. The stratified sampler achieves the best performance.

Table 10: **Performance of Different Samplers in Text-to-Image Generation Tasks.** We report the FID scores on MJHQ-30K dataset using different samplers. The proposed stratified sampler achieves the best outcome.

| Method | FID-30k ↓ |
|---|---|
| Confidence | 11.42 |
| Uniform | 8.22 |
| Halton | 7.38 |
| Stratified | **6.68** |

### B.4 ABLATION STUDIES ON DATA PIPELINE

**Effect of Task Scheduler.** To study the effect of the dataset scheduler described in Section B.1, we compare three dataset mixing strategies in Stage 1 training. The goal of Stage 1 is to equip

LaViDa with region-level understanding capabilities such as grounding. At this stage, training data includes fewer than 1M grounding samples but over 10M image-level understanding samples. To mitigate imbalance, we employ a scheduler that dynamically adjusts the sampling weights for new (grounding) and existing (image-level) capabilities. Each batch is drawn from a single dataset. For example, when New:Old=1:3, on average $\frac{1}{4}$ of batches contain grounding data and $\frac{3}{4}$ contain image-level data.

We initialize the ratio as New:Old=3:1 and gradually reduce it to 1:3. We compare against fixed ratios of 1:3 and 3:1, reporting results after 20k steps in Table 11. Fixing New:Old=1:3 under-trains grounding, while fixing New:Old=3:1 improves grounding but causes forgetting on image-level understanding. In contrast, the dynamic scheduler achieves strong performance on both. Notably, it even outperforms the fixed 3:1 setup on image-level understanding, suggesting it also mitigates overfitting caused by the small grounding dataset.

Table 11: **Comparison of different task scheduling during Stage 1 Training.** We compare the performance under different dataset sampling weights of new capabilities (grounding) and old capabilities (image-level understanding). We explored two fixed sampling ratio 1:3 and 3:1 for New:Old. For the dynamic scheduler, the New:Old ratio is initialized as 3:1 and gradually decreased to 1:3.

| Method | New Capabilities | | | Existing Capabilities | | |
|---|---|---|---|---|---|---|
| | RefCOCO | RefCOCO+ | RefCOCOg | MME | ChartQA | ScienceQA |
| New:Old = 1:3 | 83.2 | 74.6 | 78.3 | **449** | 72.6 | 84.3 |
| New:Old = 3:1 | 88.8 | 82.4 | 85.7 | 349 | 65.0 | 75.8 |
| Dynamic | **92.0** | **86.9** | **89.3** | 436 | **73.4** | **86.4** |

**Does understanding data help generation tasks?** To examine whether incorporating understanding data benefits generation, we experimented with removing all grounding data from Stage 3. The results are shown in Table 12. Even without explicit planning, incorporating grounding data enhances both text-to-image generation and editing, highlighting an inherent synergy between the tasks. When planning is enabled, these benefits compound, leading to even greater improvements.

Table 12: **Effect of Grounding Data in Stage 3 Training.** To analyze the impact of the synergy between understanding and generation tasks, we explored removing object grounding in Stage 3 Training. This leads to worse overall performance. This demonstrates that jointly training on both understanding (grounding) and generation tasks is helpful for generation.

| Method | GenEval | DPG | ImgEdit |
|---|---|---|---|
| w/o grounding data | 0.74 | 82.0 | 3.60 |
| w/ grounding data | 0.77 | 81.8 | 3.71 |
| + planning | **0.85** | **82.9** | **3.80** |

### B.5 ABLATION STUDIES ON REFLECTION AND PLANNING

**Breakdown of Performance Improvements.** We provide a detailed breakdown of the gains introduced by planning and reflection. Table 13 shows results on GenEval. Planning yields large improvements in object positioning (+0.19), while reflection additionally improves counting and attribution. To further examine the behavior of planning, we conducted additional text-to-image evaluations on categories of T2I-Compbench++ (Huang et al., 2025) that are not covered by GenEval benchmark, including 3D spatial constraints and object texture attribution. We report these results in Table 14. We observe that Lavida-O consistently demonstrate strong performance, with planning mechanism offering a significant performance boost. Notably, while our planning process use only 2D bounding boxes, we observe that it also improves satisfaction of 3D positional constraints by properly designing the size of relevant objects to reflect the distance.

On Image-Edit (Table 15), planning improves adding/removing objects, subject actions, and hybrid instructions. The largest gains are in action (+0.50) and hybrid (+0.16). However, global edits (e.g.,

style, background) degrade slightly, as these tasks are less aligned with grounding. A promising direction for future is to let the model dynamically decide whether to invoke planning.

Table 13: **Breakdown of performance improvements on GenEval Dataset.** We report the improvements of the planning and reflection mechanism on each category of the text-to-image generation tasks from GenEval Dataset.

|  | Single | Two | Position | Counting | Color | Attribution | Overall |
|---|---|---|---|---|---|---|---|
| Baseline | 0.99 | 0.85 | 0.65 | 0.71 | 0.86 | 0.58 | 0.77 |
| +Planning | 0.99 | 0.94 | 0.84 | 0.75 | 0.90 | 0.68 | 0.85 |
| $\Delta$ vs. Baseline | = | +0.09 | +0.19 | +0.04 | +0.04 | +0.10 | +0.08 |
| +Reflection | 1.00 | 0.95 | 0.89 | 0.85 | 0.90 | 0.74 | 0.89 |
| $\Delta$ vs. Baseline | +0.01 | +0.10 | +0.24 | +0.14 | +0.04 | +0.16 | +0.12 |

Table 14: **Additional Text-to-Generation results on T2I-Compbench-++ Benchmark**. We report results on categories not included in GenEval benchmark, such as 3D spatial constraints and texture attribution.

| Model | 3D | 2D | Texture |
|---|---|---|---|
| Stable Diffusion 2 (Stability AI, 2022) | 0.323 | 0.134 | 0.492 |
| Janus-Pro-7B(Chen et al., 2025c) | 0.323 | 0.157 | 0.407 |
| FLUX.1 Dev(Labs, 2024) | 0.387 | 0.286 | 0.692 |
| LaViDa-O | 0.414 | 0.388 | 0.613 |
| +Planning | **0.442** | **0.390** | **0.715** |

Table 15: **Breakdown of performance improvements on Image-Edit Dataset.** We report the improvements of the planning mechanism on each category of the image editing tasks from Image-Edit Dataset.

| Model | Add | Adjust | Extract | Replace | Remove | Background | Style | Hybrid | Action | Overall |
|---|---|---|---|---|---|---|---|---|---|---|
| Baseline | 4.04 | 3.62 | 2.01 | 4.39 | 3.98 | 4.06 | 4.82 | 2.94 | 3.54 | 3.71 |
| + Planning | 4.11 | 3.67 | 2.04 | 4.40 | 4.05 | 4.00 | 4.75 | 3.10 | 4.04 | 3.80 |
| $\Delta$ vs. Baseline | +0.07 | +0.05 | +0.03 | +0.01 | +0.07 | -0.06 | -0.07 | +0.16 | +0.50 | +0.09 |

Table 16: **Performance and Latency at different numbers of reflection rounds** $N$. When $N = 1$, we only perform planning.

| Num. of Reflection Rrounds | N=1 | N=2 | N=4 | N=8 | N=12 | N=16 | N=20 |
|---|---|---|---|---|---|---|---|
| GenEval Score $\uparrow$ | 0.848 | 0.864 | 0.875 | 0.882 | 0.890 | 0.886 | 0.886 |
| Latency (s/image) $\downarrow$ | 27.2 | 32.6 | 39.3 | 47.1 | 53.4 | 58.3 | 62.2 |

**Effect of Inference-time Scaling.** We evaluate reflection scaling by varying $N$, the maximum number of images generated per prompt. Table 16 shows results. Even one reflection step ($N = 2$) improves performance. Gains saturate at $N = 8$, with little benefit beyond. Latency grows sublinearly with $N$ since simple prompts often trigger early stopping. For example, when $N = 20$, the model may obtain a satisfactory output and terminate the generation process after generating just two images.

## B.6 ABLATION STUDIES ON UNIVERSAL TEXT CONDITIONING.

To examine the effectiveness of universal conditioning, we perform a user study and ask human evaluators to compare image generation with and without universal text conditioning. We curated a

Figure 13: **Human Evaluation of Image Quality.** We conduct user study on image quality and compare text-to-image generations with and without universal text conditioning. Results show that universal text conditioning with aesthetic scores greatly improves image quality.

total of 300 human response on image pairs generated with randomly selected prompts from MJHQ-30k dataset. The human evaluators are provided with the following instruction:

> **Instructions**
> Both of these images were generated by AI models trained to create an image from a text prompt. Which image do you prefer given the associated text?
> Example criteria could include: detail, art quality, aesthetics, how well the text prompt is reflected, lack of distortions/irregularities (e.g. extra limbs, objects). In general, choose which image you think you would consider to be "better".

We report the results in Figure 13. Results show that human evaluators exhibit a strong preference towards images generated with universal text conditioning, suggesting that conditioning the image generation with quality scores through our proposed universal text conditions method can effectively improve image quality.

## B.7 SPEED–QUALITY TRADEOFF

A key advantage of masked diffusion models over autoregressive models is the speed–quality tradeoff enabled by parallel decoding. We study this in the unified setting by evaluating Lavida-O on MJHQ-30k text-to-image generation (Li et al., 2024b), RefCOCO grounding (Kazemzadeh et al., 2014), and MathVista reasoning (Lu et al., 2023).

For MJHQ and RefCOCO, we vary the number of diffusion steps. For MathVista, we employ Fast-DLLM (Wu et al., 2025a), which adaptively unmasks multiple tokens per step. The tradeoff is controlled via its threshold hyperparameter. Results are shown in Figure 14. For MJHQ we report FID (lower is better), for RefCOCO Precision@0.5 (higher is better), and for MathVista accuracy (higher is better).

We compare against several baselines: Flux (Labs, 2024) on T2I, Qwen2.5-VL-7B (Bai et al., 2025) on grounding, and Qwen2.5-VL/Open-LLaVA-Next-8B (Chen & Xing, 2024) on reasoning. Lavida-O achieves faster inference and stronger quality on image generation and grounding. For grounding, it reaches up to $6.8\times$ speedup while surpassing Qwen2.5-VL-7B in precision. On MathVista, while less accurate than state-of-the-art AR models, Lavida-O is much faster, and still stronger than popular AR baselines such as Open-LLaVA-Next-8B. Performance also exceeds the base LaViDa (56.9 vs. 44.8).

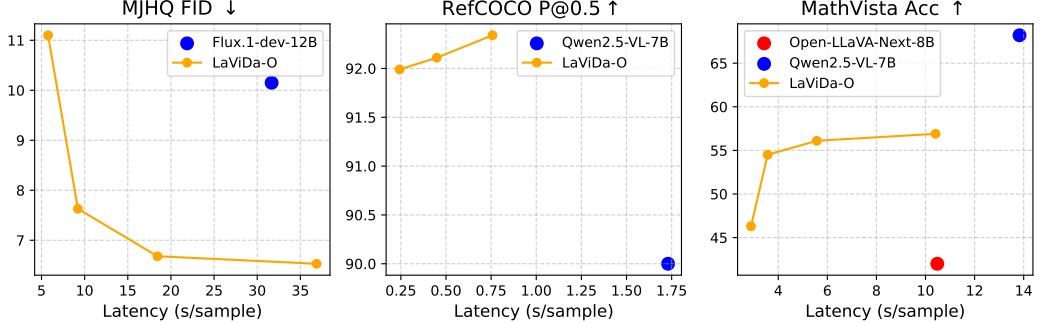

Figure 14: **Speed–quality tradeoff on generation, grounding, and reasoning.** Latency (s/sample) and benchmark scores are shown. For MJHQ: FID (lower is better). For RefCOCO: Precision@0.5 (higher is better). For MathVista: accuracy (higher is better). On MathVista, the maximum generation length is capped at 256 tokens.

### B.8 ADDITIONAL QUALITATIVE RESULTS

Finally, we provide additional qualitative examples demonstrating Lavida-O's capabilities on diverse prompts and editing instructions. Figure 15 shows text-to-image generation, and Figure 16 shows image editing results.

## C COMPUTE COST

All experiments are conducted on 8 nodes, each equipped with 8 A100 GPUs. The total training amounts to 34.2 days measured by wall clock time, or 53k GPU hours.

## D LIMITATIONS

In this section, we discuss several limitations of Lavida-O.

**Text Rendering.** Since the image generation branch is trained from scratch and we did not explicitly include datasets for text rendering, Lavida-O 's capability to render and edit text is very limited. We also find that the VQ image tokenizer we use cannot faithfully reconstruct small texts. We aim to address this issue in future work by incorporating additional text rendering data and finetune the VQ image tokenizer on screenshots of documents.

**Pixel Shift.** Our image editing datasets, such as GPT-Image-Edit-1.5M (Wang et al., 2025b) contains images distilled from generative models like GPT-4o, which is known to have "pixel shift" problems. Specifically, even if the instruction only requires editing a specific region, the other regions may still experience small but noticeable changes. As a consequence, Lavida-O inherit this problem. We aim to mitigate this by obtaining more clean and high-quality image-editing data.

**Math Reasoning.** The focus of Lavida-O is to build a unified multi-modal MDMs capable of both understanding and generation tasks. Although its math reasoning capabilities has improved from the base model LaViDa thanks to additional training, there remains a considerable gap when compared against state-of-the-art models. We leave further improvements on math reasoning tasks to future work.

**Hallucination.** Like all generative models, ours may occasionally produce inaccurate or fabricated information. We recommend using model outputs as guidance rather than unquestioned truth, and validating them where accuracy is critical.

## E    BOARDER IMPACT

Lavida-O has strong text-to-image generation capabilities and image-editing capabilities, which may be abused to create various harmful and offensive content. We strongly caution the community against such use cases. Additonally, our model may inherit the biases embedded in the base model LaViDa, as well as biases incorporated in the images and texts of the training data. Our model is intended to be used by researchers to build a strong diffusion model for multi-modal applications and explore methods of building future multi-modal foundational models. We do not recommend that it be used for any other purposes.

## F    LLM USAGE

We use LLM to correct typos and grammatical errors only.

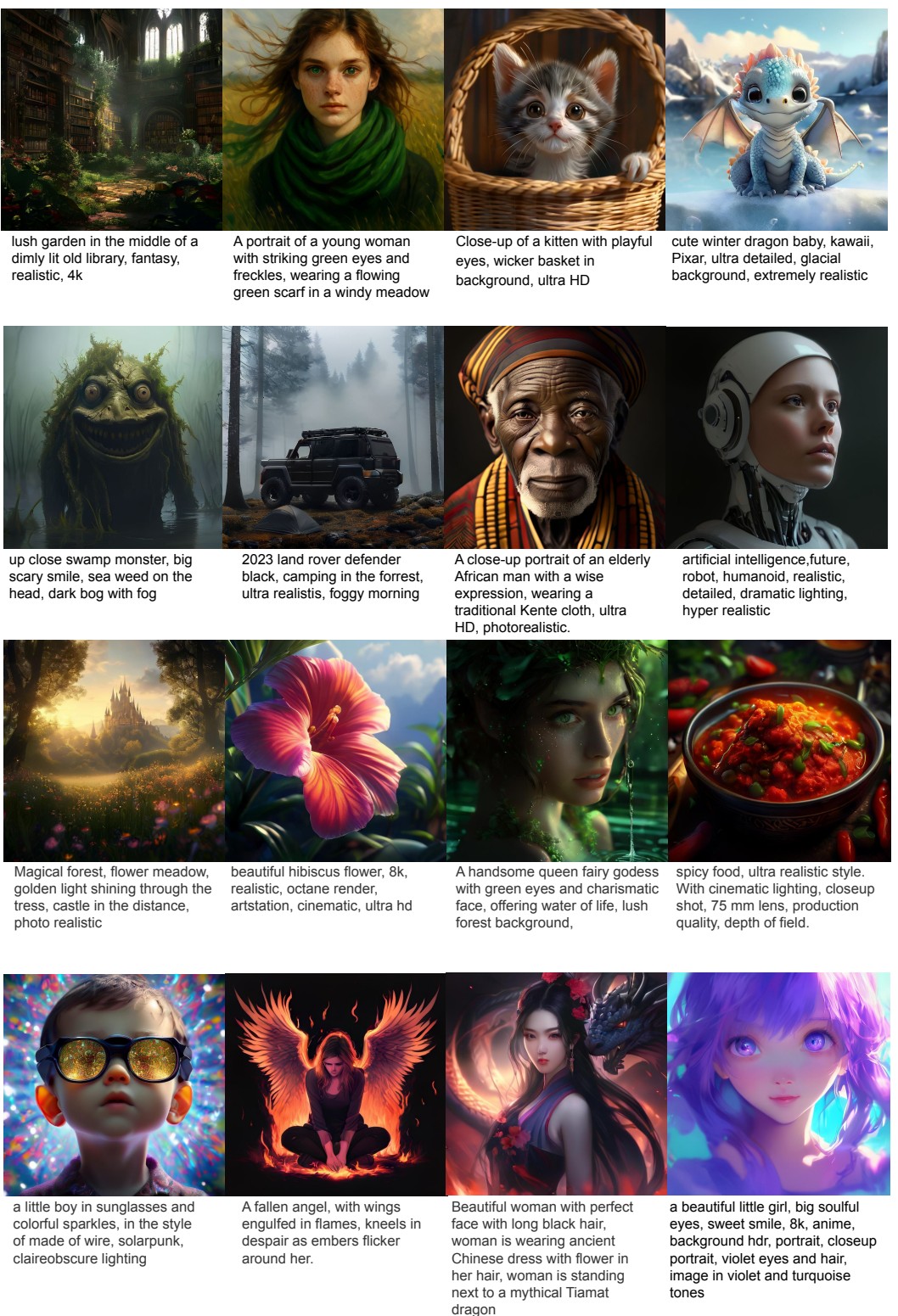

Figure 15: **Qualitative examples of text-to-image generation.** We provide additional examples of text-to-image generation outputs on diverse prompts.

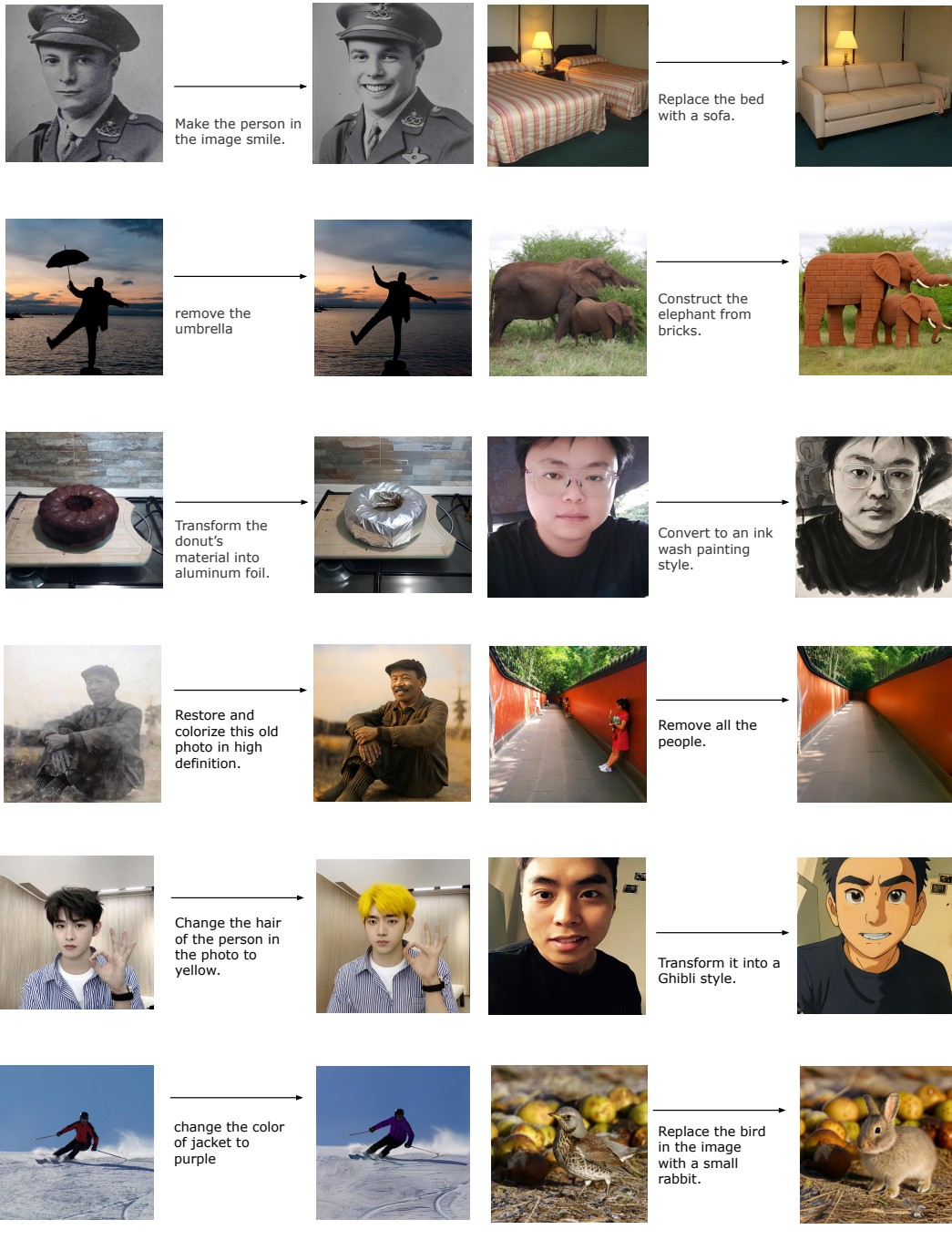

Figure 16: **Qualitative examples of image editing.** We provide additional examples of image editing outputs on diverse instructions.

