# OpenReview forum: "Lavida-O: Elastic Large Masked Diffusion Models for Unified Multimodal Understanding and Generation"
_ICLR.cc/2026/Conference — ICLR 2026 Poster_

### Official Review · Reviewer_dKrA · 2025-10-28

**Soundness:** 3
**Presentation:** 3
**Contribution:** 3
**Rating:** 6
**Confidence:** 4

**Summary:**

This paper presents Lavida-O, a unified Masked Diffusion Model for multimodal understanding and generation. Building upon LaViDa, the model integrates an Elastic Mixture-of-Transformers (Elastic-MoT) architecture with asymmetric branches for efficient parameter sharing. It supports diverse tasks such as image understanding, object grounding, image editing, and high-resolution text-to-image generation, achieving state-of-the-art performance on multiple benchmarks.

**Strengths:**

1.Proposes a unified framework that effectively bridges image understanding and generation.

2.The Elastic-MoT architecture is efficient, reducing training cost while maintaining strong performance.e

3. Demonstrates impressive empirical results across several benchmarks, indicating strong generalization.

**Weaknesses:**

1.The novelty of the work appears somewhat incremental — the Mixture-of-Transformers (MoT) paradigm has been extensively studied, and the adoption of discrete diffusion is also not particularly new. The paper’s main contribution seems to lie in engineering integration and architectural refinement, rather than proposing a fundamentally new concept.

2.The training cost appears rather high. According to the appendix, the model requires around 34 days of training, which may be excessive for a unified framework. It would be helpful if the authors could discuss potential efficiency improvements or training optimizations that make the approach more practical.

**Questions:**

see the weakness.

---

> ### Author Response · Authors · 2025-11-19
>
> # W1  This work is incremental
> While our work contains many “architectural refinement”, our work also contains many fundamentally new paradigms. Specifically, we can group our contribution in three categories.
>
> #### Category A: Novel Paradigm for Multi-modal Understanding and Generation tasks
>
> 1. **We introduce a novel paradigm of unified interleaved generation via discrete diffusion.** Unlike previous work such as MMaDa which only explored simple text-to-image generation, image understanding, or concurrent generation of image-caption pairs, our work extends the unified diffusion model to non-trivial interleaved generation tasks (planning and reflection) and demonstrated that we can leverage understanding capabilities to improve generation through interleaved reasoning in a unified paradigm.
> 2.  **We introduce a novel paradigm of multi-modal reasoning with interleaved planning and self reflection. (Sec 3.2, Appendix A7)** Previous works on unified models such as BAGEL explored improving image generation performance via rewriting prompts with chain-of-thoughts reasoning. LaViDa-O uniquely introduces a new reasoning paradigm that uses a unified model to plan with bounding boxes and reason through iterative self-reflection and feedback. We perform a thorough analysis on the effectiveness of this design (Appendix B.5), demonstrating the effectiveness of this framework on text-to-image generation and image editing tasks.
> 3. **We propose a novel paradigm for multi-object Grounding (Sec 3.2, Appendix A.5, Figure 10).** We present a novel method that formulate multi-object bounding box prediction as a text-infilling task where the bounding boxes coordinates of multiple objects can be decoded in a single step using a MDM. This formulation greatly differs from the existing QA style bounding box prediction with VLM, and demonstrated strong performance both in terms of quality and speed (Table 4, Figure 5)
> 4. **We propose a novel paradigm for conditioning a text-to-image generation model.** (Sec 3.2, Appendix A.4, Figure 8) Traditionally, microconditioning uses special embeddings to represent floating point conditions such as aesthetic score (e.g. SDXL). We first showed that with a unified model capable of numerical reasoning, one can simply encode these conditions as plain text (e.g. "Score 5.223”). This helps improves image quality (Appendix B.6)
>
> #### Category B: Novel Training and sampling techniques for discrete diffusion models
>
> 5. **We design the Modality-Aware Masking scheme that solves the interleaved generation process of multimodal MDMs. (Sec 3.1.2, Appendix A.3)** Modality-Aware Masking (Sec 3.1.2, Appendix A.3) augments the foundational formulation of masked diffusion process by making the forward and reversal process modality-aware through a special [exp] token. This distinct design of LaViDa-O uniquely addresses the challenges of interleaved generation tasks such as planning and reflection, where if/how many images are present in the model response cannot be determined in advance.
>
> 6. **We propose novel sampling method for Masked Image Generators (Sec 3.2, Appendix A.5).**  We proposed a stratified-random sampling process that greatly improves the image generation quality measured by FID score. It outperforms previous state-of-the-art Halton Sampler (ICLR 2025). (Results shown in Appendix B.3)
>
> #### Category C: Novel Architecture
>
> 7. **We solve the efficiency and scalability problem of MoT architecture  (Sec 3.1.1, Appendix A.2).** Elastic-MoT architecture challenges a fundamental assumption in MoT that image and text modalities require the same model capacity, which is not well-justified. We perform a comprehensive study in Appendix B.2. and showed that Elastic-MoT not only improves the training efficiency by 3.17× but also leads to faster convergence.
>
> # W2 The training cost appears rather high, It would be helpful if the authors could discuss potential efficiency improvements.
>
> We note that while the number is high in absolute terms, it is not significant when compared to other multi-modal foundational models. For example, Janus-Pro-7B was trained on 32 nodes for 14 days, or 3584 GPU hours. We only use 8 nodes for 34 days, which is 2176 GPU hours. While more recent works like BAGEL did not disclose the exact compute used, they train a twice larger model (14B) on more data (>5T tokens for BAGEL vs 1.3T tokens for LaViDa-O ), and is expected to have much higher compute cost.
> In terms of efficiency improvements, we highlight that one of our major contribution Elastic-MoT architecture specifically focuses on improving the training speed. We already provided a detailed analysis on the speed improvements in Appendix (B.2), which shows that it achieves up to 3.17× training speed compared to vanilla MoT used by BAGEL using the same infrastructure.

---

> > ### Comment · Reviewer_dKrA · 2025-11-28
> >
> > I have read your rebuttal carefully and I thank your response. I think it is an excellent work and I will maintain my positive attitude and score. I have no more questions.
> >
> > Best,

---

### Official Review · Reviewer_VkdP · 2025-10-28

**Soundness:** 4
**Presentation:** 4
**Contribution:** 3
**Rating:** 6
**Confidence:** 4

**Summary:**

This paper introduces Lavida-O, a unified Masked Diffusion Model (MDM) designed to handle a wide range of multimodal tasks, including image-level understanding, object grounding, image editing, and high-resolution (1024px) text-to-image synthesis. The key idea is to build a single, efficient framework that surpasses existing specialized models and other unified approaches (like AR or AR+diffusion models).

**Strengths:**

- The proposed architecture is efficient and effective. It intelligently combines a large 8B understanding model with a smaller 2.4B generation model, improving training/inference efficiency (Fig 5) while maximizing performance by leveraging the pre-trained understanding branch.

- The integration of planning (layout generation) and self-reflection (iterative refinement) is a key innovation. It provides a concrete mechanism for the model's understanding and generation capabilities to mutually benefit each other, leading to large, quantifiable gains in prompt-following and editing (Tables 3, 13, 14).

- The paper is packed with valuable technical innovations for MDMs, including stratified random sampling (improves FID, Table 10), universal text conditioning (improves control), and coordinate quantization (enables parallel decoding for grounding).

**Weaknesses:**

- The paper acknowledges that the model's text rendering capability is "very limited." This is a significant weakness for a model aiming for general-purpose multimodal capabilities, as text is a common element in user requests. The authors attribute this to the VQ tokenizer and data, but it remains a key area for improvement.
- The model suffers from a "pixel shift" problem, where non-edited regions are altered during editing. While attributed to the training data, this lack of precision detracts from the quality of the editing feature.
- While performance on MathVista improved over the base model, it still "lags behind state-of-the-art AR models" (Sec D, Fig 13). This indicates that the MDM architecture, or at least this implementation, has not yet closed the gap on high-level, abstract reasoning.

**Questions:**

- The planning and reflection mechanisms appear to be invoked via specialized prompts (e.g., "please generate a layout...", Sec A.7). Does the model have any capability to *autonomously* decide when to use planning or reflection based on its own assessment of the prompt's complexity? Or must the user always explicitly invoke these modes?
- Regarding the "pixel shift" limitation: Beyond identifying the training data as the source, have the authors explored any mitigation strategies? For example, could a mask-preservation loss (e.g., L1 loss on non-edited regions) be applied during the image editing finetuning stage?
- The modality-aware masking (Sec 3.1.2) is shown for single-turn planning/reflection. How robust is this mechanism for more complex, multi-turn interleaved tasks (e.g., generating a story with multiple images and text blocks)? What are the practical context length limits when history includes multiple VQ-tokenized images?

---

> ### Author Response · Authors · 2025-11-19
>
> # W1 VQ tokenizer has limited text rendering capabilities.
>
> Yes, we agree that VQ tokenize is a bottleneck for text rendering and is an active research area. We note that a recent work [1] (Oct 30, 2025)  was able to greatly improve the text reconstruction of VQ tokenizers by training of specialized data such as TextAtlas5M[2] and scaling the decoder. We will explore adopting these developments in future works.
>
> [1]Cui, Yufeng, et al. "Emu3. 5: Native Multimodal Models are World Learners." arXiv preprint arXiv:2510.26583 (2025).
>
> [2] Jinpeng Wang, Alex, et al. "TextAtlas5M: A Large-scale Dataset for Dense Text Image Generation." arXiv e-prints (2025): arXiv-2502.
>
> # W2 & Q2 How to address the pixel shift problems beyond identifying the training data?
> While some works use mask preservation loss like L1 regularizer, this is challenging because we represent images as discrete tokens, so we cannot back-propagate pixel-space losses back to the model. Moreover, since the image in the training data has ‘pixel-shift’ problems, the encoded token representation of input and target image will differ even in non-edited regions. In general, adopting the idea of L1 regularizer to discrete image generators is non-trivial and requires designing new losses applicable to token distributions (e.g. some form of KL penalty). We will explore this in future works.
>
> # Q1 Can the model automatically decide when to perform reason or must the user always explicitly invoke these modes?
>
> No.  In general, most pure language models also require special chain-of-though prompts to enable reasoning and planning capabilities, as opposed to directly generating an answer.  We adopt this design to give user the ability to control the tradeoff between speed and quality, by consciously enabling and disabling the planning feature.  As described in Appendix B.5, ”A promising direction for future is to let the model dynamically decide whether to invoke planning”. We will explore this adding automatic routing for planning modes in future.
>
> # Q3 How robust is this mechanism for more complex, multi-turn interleaved tasks?
>
> We want to clarify that while Sec 3.1.2 describes only single turn behavior, the reflection is a multi-turn process, and additional results at different number of round are already provided in Appendix B.5 table 16, where we report results up to 20 rounds. In practice, the max context limit of the model is 3 rounds. However, we can extend the generation beyond 3 rounds by context truncation, and only preserve the original input and three latest reflection rounds. Results show that this design leads to continued improvement in generation quality beyond 3 rounds.
>
> We agree that multi-image story generation is an interesting task, we will explore such applications in future works by including additional training data.

---

### Official Review · Reviewer_NaEt · 2025-10-31

**Soundness:** 4
**Presentation:** 4
**Contribution:** 3
**Rating:** 8
**Confidence:** 3

**Summary:**

The paper proposes LaViDa-O, a unified masked diffusion model that brings image understanding, object grounding, text-to-image generation, image editing, planning and self-reflection into one architecture. It includes several techniques, like Elastic-MoT, modality-aware masking, universal text conditioning, and stratified sampling. Empirically, the planning and reflection produces noticeable gains on text-to-image and editing benchmarks, and the model also improves understanding relative to the LaViDa base.

**Strengths:**

1. The paper is clearly written and well organized.

2. The method combines multiple ideas designed for different challenges (e.g., stratified sampling). There is a clear motivation before proposing each.

3. Elastic-MoT is a sensible design for improving efficiency, and the appendix provides informative ablations exploring its effectiveness.

4. The single model shows clear performance gains across multiple dimensions (e.g., generation and understanding), particularly when incorporating planning and reflection.

Overall, the paper introduces several clever design choices, such as modality-aware masking and stratified sampling, each of which is well-motivated and justified. So, I believe this work will positively influence the development of masked diffusion models.

**Weaknesses:**

1. Reporting results on some components of T2I-CompBench++ would strengthen the evaluation, especially since some of its focused prompts (e.g., for 3D and texture) are not currently covered by GenEval.

2. The contribution of universal text conditioning is not quantitatively assessed. A user study or automated evaluation demonstrating its benefit would be helpful.

**Questions:**

1. For editing images, it may be the case that other conditioning modalities (e.g., keypoints, masks) might be more beneficial than bounding boxes. Is there a way to extend the grounding mechanism to such modalities?

2. The paper states that bounding boxes are denoised in a single step. Could the authors clarify how this is implemented (when to unmask)? Also, is the timestep necessarily before the denoising step of an [exp] token?

3. During training, do image tokens follow a separate masking schedule until $t_{\text{exp}}$? If so, would unmasking an [exp] token before or after $t_{\text{exp}}$ introduce a train-inference mismatch?

---

> ### Author Response · Authors · 2025-11-19
>
> # W1 Reporting results on some components of T2I-CompBench++
> We thank the reviewer for the suggestion and included additional results of T2I-CompBench++ to cover the categories not included in GenEval. The new results are added to appendix B.5 (text highlighted in red). For convenience, we also included a copy as follows
>
> | **Model**                         | **3D** | **2D** | **Texture** |
> |----------------------------------|-------:|-------:|------------:|
> | Stable Diffusion 2  | 0.323 | 0.134 | 0.492 |
> | Janus-Pro-7B       | 0.323 | 0.157 | 0.407 |
> | FLUX.1 Dev            | 0.387 | 0.286 | 0.692 |
> | **LaViDa-O**                      | 0.414 | 0.388 | 0.613 |
> | **+Planning**                     | **0.442** | **0.390** | **0.715** |
>
> We observe that LaViDa-O consistently demonstrate strong performance, with planning mechanism offering a significant performance boost. Notably, while our planning process use only 2D bounding boxes, we observe that it also improves satisfaction of 3D positional constraints by properly designing the size of relevant objects to reflect the distance.
>
> # W2 A user study on Universal Text Conditioning would be helpful.
>
> We thank the reviewer for the suggestion and included the results in Appendix B.6. As requested, we perform a user study and ask human evaluators to compare image generation with and without universal text conditioning. We curated a total of 300 human response on image pairs generated with randomly selected prompts from MJHQ-30k dataset. The human evaluators are provided with the following instruction.
>
> We report the results in Figure 13 (page 30 of revision). Results show that human evaluators exhibit a strong preference towards images generated with universal text conditioning (69% prefer generation w/ universal conditioning, 5% tie, 26% prefer generation w/o universal conditioning), suggesting that conditioning the image generation with quality scores through our proposed universal text conditions method can effectively improve image quality.
>
> # Q1 Is there a way to extend the grounding mechanism to such modalities?
>
> Yes. The bounding box is represented as 4 points. To represent key points, we can simply use >4 points to represent the location of all the joints. For masks, they can be simply encoded as an extra input image, (e.g. a copy of original input image with mask overlay).  We will incorporate data from more modalities in future works.
>
> # Q2  How is bounding box decoding implemented
>
> For purely grounding tasks, the output consists of only text tokens. Bounding boxes prediction is implemented as a text-inflling task (see Figure 10) where we use a dummy prompt “please describe the locations of the objects in the image” and prefill a draft answer such as “a dog [m][m][m][m], a boy [m][m][m][m]”. For interleaved generations such as planning, recall that bounding boxes are represented as special tokens (details in Sec A.6) and each bounding box consists of exactly 4 tokens, we can simply unmask all of them when 1) the clean token prediction X_0 from model has 4 consecutive bounding box tokens, 2) and one of them will be unmasked according to the sample schedule.
>
> For planning, we defer the unmasking of image tokens until after bounding boxes have been created in the sampling process, the timestep is always larger than t_exp (recall that X_0 has only clean token, X_1 has only masked tokens)
>
> # Q3 During training, do image tokens follow a separate masking schedule?
> This is true, during the training, the image  tokens does have an independent masking rate. Effectively, we have x% of images tokens masked and y% of text tokens masked, with x, y sampled independently. This does not lead to training-inference mismatch, since this setup allows the model to predict with arbitrary combinations of masking ratios. In fact, if we want, we can even retrospectively create planning bounding boxes for an already generated image by setting the masking ratio of image tokens to 0% and that of text tokens to 100%, although this is not the standard decoding process.

---

> > ### Comment · Reviewer_NaEt · 2025-11-27
> >
> > I thank the authors for the rebuttal and for providing additional experiments and clarifications. My concerns have been addressed, and I will maintain my current score of acceptance.

---

### Official Review · Reviewer_GnXG · 2025-11-01

**Soundness:** 2
**Presentation:** 3
**Contribution:** 2
**Rating:** 4
**Confidence:** 4

**Summary:**

This paper presents a unified masked model that is able to tackle multimodal understanding and generation. Specifically, several components, including Elastic Mixture-of-Transformers, planning, and iterative self-reflection, were proposed. Extensive experimental results on understanding and generation benchmarks demonstrated the effectiveness of the proposed approach.

**Strengths:**

1. This paper is well-organized and easy to read.
2. The proposed approach demonstrates promising results on several understanding and generation tasks, such as grounding and editing.
3. Comprehensive evaluations are conducted.

**Weaknesses:**

1. The proposed masked diffusion language model is not relatively interesting, which was explored in UMDD[1].
2. The overall methodologies, including mixture-of-transformer, discrete language model, and discrete diffusion, are well-studied by the existing works [1,2,3,4,5]. I cannot find a significant contribution of this work.
3. There are several recent works with models of around 2B parameters, which should be fairly compared in the benchmarks.


[1] Unified Multimodal Discrete Diffusion.
[2] LMFusion: Adapting Pretrained Language Models for Multimodal Generation.
[3] Emerging Properties in Unified Multimodal Pretraining.
[4] MMaDA: Multimodal Large Diffusion Language Models.
[5] Show-o: One Single Transformer to Unify Multimodal Understanding and Generation.

**Questions:**

See weaknesses. I would adjust my initial rating according to the authors' response.

---

> ### Author Response · Authors · 2025-11-19
>
> ### W1
> For completeness, we have revised related works section to include UMDD discussion. However, we respectfully disagree that our work is "not relatively interesting" compared to UMDD[1]. While UMDD pioneered the concept of unified discrete diffusion, Lavida-O introduces specific architectural and paradigmatic innovations required to make this approach scalable and capable of complex reasoning. We highlight four key differentiators:
>
> **Unified Generation Paradigm.** Unlike UMDD which only explored simple text-to-image generation, image understanding, or concurrent generation of image-caption pairs [1], our work extends the unified diffusion model to non-trivial interleaved generation tasks (planning and reflection) and demonstrated that we can leverage understanding capabilities to improve generation through interleaved reasoning in a unified paradigm.
>
> **Task Coverage.** Unlike prior works which support limited tasks, LaViDa-O is the first masked diffusion model to achieve 1024px high resolution image generation, localized understanding and object grounding, and instruction-based image-editing. (Table 1) Further, it achieves state-of-the-art performance on all these tasks that are not previously explored by masked diffusion models, demonstrating their great potential.
>
> **Architectural Innovation:** UMDD relies on a standard dense Transformer architecture. As noted in (Sec 3.1.1), scaling dense unified models is computationally challenging. Lavida-O introduces the Elastic Mixture-of-Transformers (Elastic-MoT). Unlike standard MoTs (e.g., BAGEL) that simply double parameters, Elastic-MoT couples a lightweight generation branch (2.4B) with a larger understanding branch and utilizes decoupled attention in later layers. This novel design allows us to activate only 6.4B parameters for generation tasks while maintaining the performance of an 8B+ model, a critical scaling solution absent in UMDD.
>
> **Scale and performance** Unlike previous works with limited scale and subpar performance, LaViDa-O is the first open-sourced, open-data multi-modal discrete diffusion model that achieves state-of-the-art performance on a wide range of generation tasks including text-to-image generation and image editing. It outperforms frontier open-sourced T2I diffusion models like Flux and unified understanding and generation models like BAGEL, demonstrating that masked diffusion models are a viable choice to build frontier foundational models. We argue that these results are meaningful to the community and our released model will significantly advance future research in this area.

---

> > ### Author Response · Authors · 2025-11-19
> >
> > ### W2
> > We respectfully disagree with the assessment that our methodologies are merely "well-studied" applications of existing works.  While it may be true that mixture of transformers,  discrete diffusion, and unified models have been explored, we emphasize that they are still active research areas with many new works focusing on improving MoT architecture, the training and sampling of discrete diffusion models, etc .  While we build upon foundational concepts like MoT and Discrete Diffusion, Lavida-O introduces fundamental architectural and algorithmic innovations designed specifically to solve the efficiency and quality bottlenecks that standard versions of these methods face at scale. In general, our contribution comes in three categories:
> >
> >
> > #### Category A: Novel Architecture:
> >
> > 1. **We solve the efficiency and scalability problem of MoT architecture  (Sec 3.1.1, Appendix A.2).** Elastic-MoT architecture challenges a fundamental assumption in MoT that image and text modalities require the same model capacity, which is not well-justified. We perform a comprehensive study in Appendix B.2. and showed that Elastic-MoT not only improves the training efficiency by 3.17× but also leads to faster convergence.
> >
> > #### Category B: Novel Training and sampling techniques for discrete diffusion models:
> >
> > 2. **We design the Modality-Aware Masking scheme that solves the interleaved generation process of multimodal MDMs. (Sec 3.1.2, Appendix A.3)** Modality-Aware Masking (Sec 3.1.2, Appendix A.3) augments the foundational formulation of masked diffusion process by making the forward and reversal process modality-aware through a special [exp] token. This distinct design of LaViDa-O uniquely addresses the challenges of interleaved generation tasks such as planning and reflection, where if/how many images are present in the model response cannot be determined in advance.
> >
> > 3. **We propose novel sampling method for Masked Image Generators (Sec 3.2, Appendix A.5).**  We proposed a stratified-random sampling process that greatly improves the image generation quality measured by FID score. It outperforms previous state-of-the-art Halton Sampler (ICLR 2025). (Results shown in Appendix B.3)
> >
> > #### Category C: Novel Paradigm for Multi-modal Understanding and Generation tasks
> >
> > 4. **We introduce a novel paradigm of multi-modal reasoning with interleaved planning and self reflection. (Sec 3.2, Appendix A7)** Previous works on unified models such as BAGEL explored improving image generation performance via rewriting prompts with chain-of-thoughts reasoning. LaViDa-O uniquely introduces a new reasoning paradigm that uses a unified model to plan with bounding boxes and reason through iterative self-reflection and feedback. We perform a thorough analysis on the effectiveness of this design (Appendix B.5), demonstrating the effectiveness of this framework on text-to-image generation and image editing tasks.
> >
> > 5. **We propose a novel paradigm for multi-object Grounding (Sec 3.2, Appendix A.5, Figure 10).** We present a novel method that formulate multi-object bounding box prediction as a text-infilling task where the bounding boxes coordinates of multiple objects can be decoded in a single step using a MDM. This formulation greatly differs from the existing QA style bounding box prediction with VLM, and demonstrated strong performance both in terms of quality and speed (Table 4, Figure 5)
> >
> > 6. **We propose a novel paradigm for conditioning a text-to-image generation model.(Sec 3.2, Appendix A.4, Figure 8)** Traditionally, microconditioning uses special embeddings to represent floating point conditions such as aesthetic score (e.g. SDXL). We first showed that with a unified model capable of numerical reasoning, one can simply encode these conditions as plain text (e.g. "Score 5.223”). This helps improves image quality (Appendix B.6)
> >
> > We highlight that these individual contributions are accompanied by comprehensive analysis. Their **scope and significance is comparable to specialized focused works** (e.g. halton scheduler[2], accepted to ICLR 2025, focus solely on proposing a sampling method for masked image generators). Due to space constraints, we offer high level descriptions  in Sec 3.2 of the main paper and leave more thorough experiment results in the appendix.

---

> ### Author Response · Authors · 2025-11-19
>
> ### W3
>
> We respectfully appreciate the suggestion to ensure fair comparisons. We would like to clarify our experimental setup and the rationale behind our baselines.
>
> 1. **Inclusion of ~2B Baselines:** We explicitly included models in the ~1-2B parameter range in Table 3 for completeness, specifically SD3-Medium (2B), Show-o (1.3B), Meissonic (1B), and Muddit (1B). Lavida-O outperforms these baselines on text-to-image benchmarks (e.g., 0.77 GenEval vs. SD3-Medium's 0.74).
>
> 2. **Fairness of the Comparison:** While Lavida-O has a larger total parameter count (10.4B active), we argue the comparison with 2B models is fair regarding generative capacity becasue. Specifically, for SD3 comparison, the pixel synthesis is handled by this 2.4B generation branch, which has similar capacity to SD3's DiT architecture. We note that the SD3 parameter count does not include its 4.7B text encoder, and their total parameter count is 4.7B+2B, which is comparable to our 4B+2.4B setup (Figure 3). Furthermore, establishing an identical experimental setup is inherently challenging, as foundational baselines like Flux, Bagel, and SD3 rely on non-public, proprietary training data.
>
> 3. **Necessity of Scale for Unified Tasks:** The large parameter count in our model are necessary to bridge the gap between pre-existing MDMs and state-of-the-art unified models like BAGEL (14B). A 2B total model would lack the capacity for the complex reasoning, planning, and object grounding tasks that are central to our contribution. Our design choices, such as Elastic-MoT, specifically focus on addressing these problems at scale efficiently.
>
> ### Question to Reviewer:
>
> Regarding the suggestion to compare with "several recent works at 2B scale", could you please clarify if there are specific 2B models other than those included in Table 3 you had in mind that are missing? We would be eager to include them to ensure our evaluation is comprehensive
>
> [1] Swerdlow, Alexander, et al. "Unified multimodal discrete diffusion." arXiv preprint arXiv:2503.20853 (2025).
>
> [2] Besnier, Victor, et al. "Halton scheduler for masked generative image transformer." ICLR 2025.

---

> ### Comment · Reviewer_GnXG · 2025-11-28
>
> Thank you to the authors for the comprehensive and detailed response. Overall, from my own perspective, the paper is not relatively interesting to me because the “unified discrete diffusion model” is not a fundamentally new concept and has already been explored in many prior works such as UniDisc, MMaDA, Muddit, etc. The main contributions appear to be more like a set of practical complementary techniques and tasks, and reviewer dKrA also raised a similar concern regarding incremental contribution.
>
> In other perspectives, as mentioned earlier in the “questions” section, I would be willing to adjust my initial rating because I appreciate the efforts in integrating new techniques and extensive and comprehensive experiments demonstrating the potential of unified multimodal models on more diverse tasks such as localized understanding and grounding, and especially interleaved reasoning and planning, which are likely not covered in previous literature and show the differences. Given that these contributions are primarily technical, **open-sourcing the model and the whole codebase—including training and evaluation (as the author claimed, “LaViDa-O is the first open-sourced, open-data multi-modal discrete diffusion model”)—is critical to ensure and enlarge the impact, which is meaningful for the community for reproduction**. The detailed response also enhances the necessity and technical contribution.
>
> Additionally, for unified multimodal models around the same ~2B parameter scale, such as Janus-pro-1.5B, Harmon-1.5B, UniGen-1.5B, etc., these baselines should be properly included in the main comparison tables 1 and 2. Since this work positions itself as a general unified multimodal model, comparison with these representative models is necessary to convincingly demonstrate the superiority of the proposed approach.

---

> ### Author Response · Authors · 2025-11-28
>
> Hi
> We thank the reviewer for "willing to adjust [the] scores."
> ### 2B comparisons
> We have added the required comparison for Jansus and UniGen in Table 2 and 3 of the revision.  Note that for UniGen, we report results of SFT stage for a fair comparison because they directly use GenEval metric as a reward in RL stage. We aslo note that even their RL+Chain-of-Thought results (0.85) is still worse than LaViDa-O (0.89) on GenEval.
>
> ### Open source
> We reaffirm the claim "LaViDa-O s the first open-sourced, open-data multi-modal discrete diffusion model". We thank the reviewer for recognizing the significance, and that it's "meaningful for the community".  We cannot include an anonymous link or zip in the supplementary material here because the legal policy requires us to incorporate a specific license and copyright headers in all copies of the code, which contains identifiable information.
>
> To formalize the claim, we have included a reproducibility statement (Appendix G) in the revision, which clearly states that links to model weight, training code and inference code will be added in the final version.

---

### Meta-Review · Area_Chair_VsoK · 2025-12-27

**Summary:**

As two reviewers have stated that their concerns have been addressed, the following analysis focuses on the remaining concerns raised by the other two reviewers.

`Reviewer GnXG`’s concerns mainly relate to:
1. The positioning of the contributions relative to prior work, including mixture-of-transformer models, discrete language models, and discrete diffusion approaches.
2. The absence of a 2B-parameter baseline in the benchmark evaluations.

`Reviewer VkdP`’s concerns focus on:
1. Limitations in text generation and rendering capabilities.
2. The “pixel shift” issue, where non-edited regions are altered during editing.
3. Clarification of the reasoning model and its robustness for complex, multi-turn interleaved tasks.

**Reviewer Concerns:**

For `Reviewer GnXG`, the concerns appear to be addressed, and the reviewer indicated an intention to adjust their score. The authors added the previously missing 2B-parameter baseline comparison. Although the reviewer does not consider the core concept entirely new, they acknowledged and appreciated the proposed practical, complementary techniques and tasks. In particular, the reviewer highlighted the authors’ commitment to open-sourcing the model and the full codebase, including training and evaluation, which played a significant role in their assessment.

For `Reviewer VkdP`, the questions are addressed with reasonable explanations, and the authors discuss potential directions for future work to mitigate the identified limitations.

**Reviewer Scores:**

Below are the extrapolated scores assuming a full discussion was carried out:

- `Reviewer GnXG` (initial rating: 4) is likely to increase their rating to 6 (borderline accept) or higher based on their response.
- `Reviewer NaEt` (initial rating: 8) is expected to maintain their positive assessment.
- `Reviewer VkdP` (initial rating: 6) is likely to remain at 6 or potentially increase their rating.
- `Reviewer dKrA` (initial rating: 6) is expected to maintain a positive rating

---

### Decision · Program_Chairs · 2026-01-26

Accept (Poster)